# Action Noise in Off-Policy Deep Reinforcement Learning: Impact on Exploration and Performance

**Jakob Hollenstein**                                    *jakob.hollenstein@uibk.ac.at*
*Department of Computer Science, University of Innsbruck*

**Sayantan Auddy**                                       *sayantan.auddy@uibk.ac.at*
*Department of Computer Science, University of Innsbruck*

**Matteo Saveriano**                                     *matteo.saveriano@unitn.it*
*Department of Industrial Engineering, University of Trento*

**Erwan Renaudo**                                        *erwan.renaudo@uibk.ac.at*
*Department of Computer Science, University of Innsbruck*

**Justus Piater**                                        *justus.piater@uibk.ac.at*
*Department of Computer Science, University of Innsbruck*

**Reviewed on OpenReview:** *https://openreview.net/forum?id=NljBlZ6hmG*

## Abstract

Many Deep Reinforcement Learning (D-RL) algorithms rely on simple forms of exploration such as the additive action noise often used in continuous control domains. Typically, the scaling factor of this action noise is chosen as a hyper-parameter and is kept constant during training. In this paper, we focus on action noise in off-policy deep reinforcement learning for continuous control. We analyze how the learned policy is impacted by the noise type, noise scale, and impact scaling factor reduction schedule. We consider the two most prominent types of action noise, Gaussian and Ornstein-Uhlenbeck noise, and perform a vast experimental campaign by systematically varying the noise type and scale parameter, and by measuring variables of interest like the expected return of the policy and the state-space coverage during exploration. For the latter, we propose a novel state-space coverage measure $X_{\mathcal{U}\text{rel}}$ that is more robust to estimation artifacts caused by points close to the state-space boundary than previously-proposed measures. Larger noise scales generally increase state-space coverage. However, we found that increasing the space coverage using a larger noise scale is often not beneficial. On the contrary, reducing the noise scale over the training process reduces the variance and generally improves the learning performance. We conclude that the best noise type and scale are environment dependent, and based on our observations derive heuristic rules for guiding the choice of the action noise as a starting point for further optimization. https://github.com/jkbjh/code-action_noise_in_off-policy_d-rl

## 1 Introduction

In (deep) reinforcement learning an agent aims to learn a policy to act optimally based on data it collects by interacting with the environment. In order to learn a well performing policy, data (i.e. state-action-reward sequences) of sufficiently good behavior need to be collected. A simple and very common method to discover better data is to induce variation in the data collection by adding noise to the action selection process. Through this variation, the agent will try a wide range of action sequences and eventually discover useful information.

A motivating example: Mountain-Car

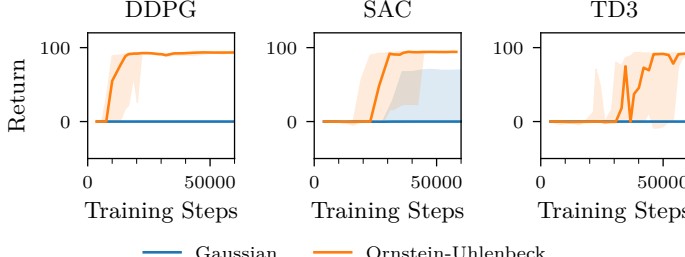

| noise Type | Gaussian | Ornstein-Uhlenbeck |
|---|---|---|
| Scale | 0.6 | 0.5 |
| Return | -30.2 | -30.4 |
| +- | 0.1 | 1.3 |

Table 1: Untrained random policies, Gaussian ($\sigma = 0.6$) and Ornstein-Uhlenbeck ($\sigma = 0.5$) achieve similar returns and appear interchangeable.

Figure 1: Training with the action noises (Table 1) shows the impact of noise type; Ornstein-Uhlenbeck solves the task, but Gaussian does not. Other algorithm parameters are taken from the tuned parameters found by Raffin (2020). The lines indicate the medians, the shaded areas the quartiles of ten independent runs.

**Action Noise** In off-policy reinforcement learning algorithms applied to continuous control domains, a go-to approach is to add a randomly-sampled *action noise* to the action chosen by the policy. Typically the action noise is sampled from a Gaussian distribution or an Ornstein-Uhlenbeck process, either because algorithms are proposed using these noise types (Fujimoto et al., 2018; Lillicrap et al., 2016), or because these two types are provided by reinforcement learning implementations (Liang et al., 2018; Raffin et al., 2021a; Fujita et al., 2021; Seno & Imai, 2021). While adding action noise is simple, widely used, and surprisingly effective, the impact of action noise type or scale does not feature very prominently in the reinforcement learning literature. However, the action noise can have a huge impact on the learning performance as the following example shows.

**A motivating example:** Consider the case of the *Mountain-Car* (Brockman et al., 2016; Moore, 1990) environment. In this task, a car starts in a valley between mountains on the left and right and does not have sufficient power to simply drive up the mountain. It needs repetitive swings to increase its potential and kinetic energy to finally make it up to the top of the mountain on the right side. The actions apply a force to the car and incur a cost that is quadratic to the amount of force, while reaching the goal yields a final reward of 100. This parameterization implies a local optimum: not performing any action and achieving a return of zero.

Driving the environment with purely random policies based on the two noise types (Gaussian, $\sigma = 0.6$, Ornstein-Uhlenbeck $\sigma = 0.5$, see Table 1), yields similar returns. However, when we apply the algorithms DDPG, TD3 and SAC (Lillicrap et al., 2016; Fujimoto et al., 2018; Haarnoja et al., 2019) to this task, the resulting learning curves (Figure 1) very clearly depict the huge impact the noise configuration has. While returns of the purely random noise-only policies were similar, we achieve substantially different learning results. Learning either fails (Gaussian) or leads to success (Ornstein-Uhlenbeck). This shows the huge importance of the action noise configuration. See Section A for further details.

**Exploration Schedule** A very common strategy in Q-learning algorithms applied to discrete control is to select a random action with a certain probability $\varepsilon$. In this *epsilon-greedy* strategy, the probability $\varepsilon$ is often chosen higher in the beginning of the training process and reduced to a smaller value over course of the training. Although very common in Q-learning, a comparable strategy has not received a lot of attention for action noise in continuous control. The descriptions of the most prominent algorithms using action noise, namely DDPG (Lillicrap et al., 2016) and TD3 (Fujimoto et al., 2018), do not mention changing the noise over the training process. Another prominent algorithm, SAC (Haarnoja et al., 2019), adapts the noise to an entropy target. The entropy target, however, is kept constant over the training process. In many cases the optimal policy would be deterministic, but the agent has to behave with similar average action-entropy no matter whether the optimal policy has been found or not.

An indication that reducing the randomness over the training process has received little attention is that only very few reinforcement learning implementations, e.g., RLlib (Liang et al., 2018), implement reducing the impact of action noise over the training progress. Some libraries, like `coach` (Caspi et al., 2017), only implement a form of continuous epsilon greedy: sampling the action noise from a uniform distribution with probability $\varepsilon$. The majority of available implementations, including `stable-baselines` (Raffin et al., 2021a), `PFRL` (Fujita et al., 2021), `acme` (Hoffman et al., 2020), and `d3rlpy` (Seno & Imai, 2021), do not implement any strategies to reduce the impact of action noise over the training progress.

Exploration schedules for action noise are also not mentioned in several recent surveys (Yang et al., 2022; Ladosz et al., 2022; Amin et al., 2021)

**Contributions**

In this paper we analyze the impact of Gaussian and Ornstein-Uhlenbeck noise on the learning process of DDPG, TD3, SAC and a deterministic SAC variant. Evaluation is performed on multiple popular environments (Table E.1): Mountain-Car (Brockman et al., 2016) environment from the OpenAI Gym, Inverted-Pendulum-Swingup, Reacher, Hopper, Walker2D and Half-Cheetah environments implemented using PyBullet (Coumans & Bai, 2016–2021; Ellenberger, 2018).

- We investigate the relation between exploratory state-space coverage $X$, returns collected by the exploratory policy $R$ and learned policy performance $P$.

- We propose to assess the state-space coverage using our novel measure $X_{\mathcal{U}\mathrm{rel}}$ that is more robust to approximation artifacts on bounded spaces compared to previously proposed measures.

- We perform a vast experimental study and investigate the question whether one of the two noise types is generally preferable *(Q1)*, whether a specific scale should be used *(Q2)*, whether there is any benefit to reducing the scale over the training progress (linearly, logistically) compared to keeping it constant *(Q3)*, and which of the parameters noise type, noise scale and scheduler is most important *(Q4)*.

- We provide a set of heuristics derived from our results to guide the selection of initial action noise configurations.

**Findings** We found that the noise configuration, noise type and noise scale, have an important impact and can be necessary for learning (e.g. Mountain-Car) or can break learning (e.g. Hopper). Larger noise scales tend to increase state-space coverage, but for the majority of our investigated environments increasing the state-space coverage is not beneficial: increased state-space coverage was associated with a reduction in performance. This indicates that in these environments, local exploration, which is associated with smaller state-space coverage, tends to be favorable. *We recommend to select and tune action noise based on the reward and dynamics structure on a per-environment basis.*

We found that across noise configurations, decaying the impact of action noise tends to work better than keeping the impact constant, in both reducing the variance across seeds and improving the learned policy performance and can thus make the algorithms more robust to the action noise hyper-parameters scale and type. *We recommend to reduce the action noise scaling factor over the training time.*

We found that for all environments investigated in this study noise scale $\sigma$ is the most important parameter, and some environments (e.g. Mountain-Car) benefit from larger noise scales, while other environments require very small scales (e.g. Walker2D). *We recommend to assess an environment's action noise scale preference first.*

## 2 Related Work

By combining Deep Learning with Reinforcement Learning in their DQN method, Mnih et al. (2015) achieved substantial improvements on the Atari Games RL benchmarks (Bellemare et al., 2013) and sparked lasting interest in *Deep Reinforcement learning* (D-RL).

**Robotic environments:** In robotics, the interest in Deep Reinforcement Learning has also been rising and common benchmarks are provided by OpenAI Gym (Brockman et al., 2016), which includes control classics such as the Mountain-Car environment (Moore, 1990) as well as more complicated (robotics) tasks based on the Mujoco simulator (Todorov et al., 2012). Another common benchmark is the DM Control Suite (Tassa et al., 2018), also based on Mujoco. While Mujoco has seen widespread adoption it was, until recently, not freely available. A second popular simulation engine, that has been freely available, is the Bullet simulation engine (Coumans & Bai, 2016–2021) and very similar benchmark environments are also available for the Bullet engine (Coumans & Bai, 2016–2021; Ellenberger, 2018).

**Continuous Control:** While the Atari games feature large and (approximately) continuous observation spaces, their action spaces are discrete and relatively small, making Q-learning a viable option. In contrast, typical robotics tasks require *continuous action spaces*, implying uncountably many different actions.

A tabular Q-learning approach or a discrete Q-function output for each action are therefore not possible and maximizing the action over a learned function approximator for $Q(s, a)$ is computationally expensive (although not impossible as Kalashnikov et al. (2018) have shown). Therefore, in continuous action spaces, *policy search* is employed, to directly optimize a function approximator *policy*, mapping from state to best performing action (Williams, 1992). To still reap the benefits of reduced sample complexity of TD-methods, policy search is often combined with learning a value function, a *critic*, leading to an *actor-critic* approach (Sutton et al., 1999).

**On- and Off-policy:** Current state of the art D-RL algorithms consist of *on-policy* methods, such as TRPO (Schulman et al., 2015) or PPO (Schulman et al., 2017), and *off-policy* methods, such as DDPG (Lillicrap et al., 2016), TD3 (Fujimoto et al., 2018) and SAC (Haarnoja et al., 2019). While the on-policy methods optimize the next iteration of the policy with respect to the data collected by the current iteration, off-policy methods are, apart from stability issues and requirements on the samples, able to improve policy performance based on data collected by *any arbitrary* policy and thus can also re-use older samples.

To improve the policy, variation (*exploration*) in the collected data is necessary. The most common form of exploration is based on randomness: in on-policy methods this comes from a *stochastic policy* (TRPO, PPO), while in the off-policy case it is possible to use a stochastic policy (SAC) or, to use a *deterministic policy* (Silver et al., 2014) with added *action noise* (DDPG, TD3). Since off-policy algorithms can learn from data collected by other policies, it is also possible to combine stochastic policies (e.g. SAC) with action noise.

**State-Space Coverage:** Often, the reward is associated with reaching certain areas in the state-space. Thus, in many cases, *exploration* is related to *state-space coverage*. An intuitive method to calculate state space coverage is based on binning the state-space and counting the percentage of non-empty bins. Since this requires exponentially many points as the dimensionality increases, other measures are necessary. Zhan et al. (2019) propose to measure state coverage by drawing a bounding box around the collected data and measuring the means of the side-lengths, or by measuring the sum of the eigenvalues of the estimated covariance matrix of the collected data. However, so far, there is no common and widely adopted approach.

**Methods of Exploration:** The architecture for the stochastic policy in SAC (Haarnoja et al., 2019) consists of a neural network parameterizing a Gaussian distribution, which is used to sample actions and estimate action-likelihoods. A similar stochastic policy architecture is also used in TRPO (Schulman et al., 2015) and PPO (Schulman et al., 2017). While this is the most commonly used type of distribution, more complicated parameterized stochastic policy distributions based on normalizing flows have been proposed (Mazoure et al., 2020; Ward et al., 2019). In the case of action noise, the noise processes are not limited to uncorrelated Gaussian (e.g. TD3) and temporally correlated Ornstein-Uhlenbeck noise (e.g. DDPG): a whole family of action noise types is available under the name of colored noise, which has been successfully used to improve the Cross-Entropy-Method (Pinneri et al., 2020). A quite different type of random exploration are the parameter space exploration methods (Mania et al., 2018; Plappert et al., 2018), where noise is not applied to the resulting action, but instead, the parameters of the policy are varied. As a somewhat intermediate method, state dependent exploration (Raffin et al., 2021b) has been proposed, where action noise is deterministically generated by a function based on the state. Here, the function parameters are changed randomly for each episode, leading to different deterministic "action noise" for each episode. Presumably

among the most intricate methods to generate exploration are the methods that train a policy to achieve exploratory behavior by rewarding exploratory actions (Burda et al., 2019; Tang et al., 2017; Mutti et al., 2020; Hong et al., 2018; Pong et al., 2020). Another alternative can be a two-step approach, where in the first stage intrinsically-motivated exploration is used to populate the replay buffer, and in the second stage the information in the buffer is exploited to learn a policy (Colas et al., 2018).

It is however, not clear yet, which exploration method is most beneficial, and when a more complicated method is actually worth the additional computational cost and complexity. In this work we aim to reduce this gap, by investigating the most widely used baseline method in more detail: exploration by action noise.

**Studies of Random Exploration** Exploration in Deep Reinforcement Learning is also the subject of multiple surveys. However, the topic of action noise is only covered very sparsely. Yang et al. (2022) only briefly mention action noise as being used in DDPG (Lillicrap et al., 2016) and TD3 (Fujimoto et al., 2018) but do not provide further discussion. A section on randomness-based methods that focuses mostly on discrete action spaces, or for continuous action spaces on parameter noise, is provided by Ladosz et al. (2022). Amin et al. (2021) provide a section on randomized action selection in policy search, nicely divided into action-space and parameter-space methods, and discuss temporally correlated or uncorrelated perturbations. However, they also do not point to any empirical study specifically comparing the effects of random exploration.

Generally, however, it appears that most work focuses on proposing modifications of the action noise, rather than investigating the effects of the baseline parameters. For example, for stochastic policies, Rao et al. (2020) show that the weight initialisation procedure can lead to different initial action distributions of the stochastic policies. Chou et al. (2017) propose stochastic policies based on the $\beta$-distribution. Nobakht & Liu (2022) use gathered experience to tune the action noise model.

In contrast to these works, in our previous work (Hollenstein et al., 2021), we investigated action noise as the immediate means to control the environments, i.e. adding action noise to a constant-zero policy. Not surprisingly, we found that there are dependencies between environment dynamics, reward structure and action noise. However, this study did not investigate the influence of action noise on learning progress or results. In this work, we investigate the impact of action noise in the context of learning.

## 3 Methods

In this section, we describe the action noise types, the schedulers to reduce the scaling factor of the action noise over time and the evaluation process in more detail. We briefly list the analyzed benchmark environments and their most important properties. We chose environments of increasing complexity that model widely used benchmark tasks. We list the used algorithms and then describe how we gather evaluation data and how it is aggregated. Last, we describe the methods we use for analyzing state-space coverage.

### 3.1 Noise types: Gaussian and Ornstein-Uhlenbeck

The action noise $\varepsilon_{a_t}$ is added to the action drawn from the policy:

$$a_t = \operatorname*{clip}_{a_{\min}, a_{\max}} \left[ \tilde{a}_t \;+\; \beta \left( \operatorname*{clip}_{-1;1}[\varepsilon_{a_t}] \cdot \frac{a_{\max} - a_{\min}}{2} + \frac{a_{\max} + a_{\min}}{2} \right) \right] \tag{1}$$

where $\tilde{a}_t \sim \pi_\theta(\cdot|s_t)$ for stochastic policies or $\tilde{a}_t = \pi_\theta(s_t)$ for deterministic policies. We introduce an additional impact scaling factor $\beta$, which is typically kept constant at the value one. In Section 3.2 we describe how we change $\beta$ over time to create a noise scheduler. The action noise $\varepsilon_{a_t}$ is drawn from either a Gaussian distribution or an Ornstein-Uhlenbeck (OU) process. The noise distributions are factorized, i.e. noise samples are drawn independently for each action dimension. For the generation of action noise samples, the action space is assumed to be limited to $[-1, 1]$ but then rescaled to the actual limits defined by the environment.

**Gaussian noise** is temporally uncorrelated and is typically applied on symmetric action spaces (Hill et al., 2018; Raffin, 2020) with commonly used values of $\mu = 0$ and $\sigma = 0.1$ with $\Sigma = \boldsymbol{I} \cdot \sigma$. Action noise is sampled

according to

$$\varepsilon_{a_t} \sim \mathcal{N}(\mu, \Sigma) \tag{2}$$

In this setup, Gaussian action noise is sampled and clipped to the action limits as needed.[1]

**Ornstein-Uhlenbeck noise** is sampled from the following temporal process, with each action dimension calculated independently of the other dimensions:

$$\varepsilon_{a_t} = \varepsilon_{a_{t-1}} + \theta(\mu - \varepsilon_{a_{t-1}}) \cdot \mathrm{d}t + \sigma\sqrt{\mathrm{d}t} \cdot \epsilon_t \tag{3}$$

$$\varepsilon_{a_0} = \mathbf{0} \qquad \epsilon_t \sim \mathcal{N}(\mathbf{0}, \mathbf{I}) \tag{4}$$

The process was originally described by Uhlenbeck & Ornstein (1930) and applied to reinforcement learning in DDPG by Lillicrap et al. (2016). The parameters we use for the Ornstein-Uhlenbeck noise are taken from a widely used RL-algorithm implementation (Hill et al., 2018): $\theta = 0.15$, $\mathrm{d}t = 0.01$, $\mu = 0$, $\sigma = 0.1 \cdot \mathbf{I}$.

Due to the huge number of possible combinations of environments, algorithms, noise type, noise scale and the necessary repetition with different seeds, we had to limit the number of investigated scales. We set out with two noise scales $\sigma$ encountered in pre-tuned hyper-parameterization (Raffin, 2020), $0.1, 0.5$, and continued with a linear increase, $0.9, 1.3, 1.7$. Much smaller noise scales vanish in the variations induced by learning and much larger scales lead to Bernoulli trials of the min-max actions without much difference.

Because the action noise is clipped to $[-1, 1]$ before being scaled to the actual action limits, a very large scale, such as $1.7$, implies a larger percentage of on-the-boundary action noise samples and is thus more similar to bang-bang control actions, the latter having been found surprisingly effective in many RL benchmarks (Seyde et al., 2021).

### 3.2 Scheduling strategies to reduce action noise

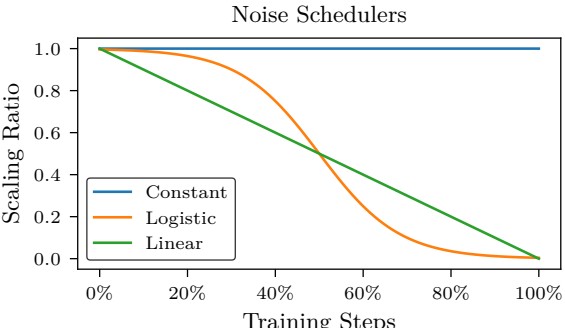

Figure 2: Action noise is used for exploration. The agent should favor exploration in the beginning but later favor exploitation. Similar to $\epsilon$-greedy strategies in discrete-action Q-learning, the logistic and linear schedulers reduce the impact of noise (scaling ratio, $\beta$ in (1)) over the course of the training progress.

In (1) we introduce the action noise scaling-ratio $\beta$. In this work we compare a constant-, linear- and logistic-scheduler for the value of $\beta$. The effective scaling of the action noise by the noise schedulers is illustrated in Figure 2. The noise types are described in more detail in Section 3.1.

Changing the $\sigma$ (see (3) and (2)) instead of $\beta$ could result in a different shape of the distribution, for example when values are clipped, or when the $\sigma$ indirectly affects the result as in the Ornstein-Uhlenbeck process. To keep the action noise distribution shape constant, the action noise schedulers do not change the $\sigma$ parameter of the noise process but instead scale down the resulting sampled action noise values by changing

---

[1]An alternative way of sampling Gaussian action noise would be to use a truncated Gaussian distribution. We investigate non-truncated Gaussian distributions together with clipping, as they are more common in practice.

the $\beta$ parameter: this means that the effective range of the action noise, before scaling and adjusting to the environment limits, changes over time from $[-1, 1]$, the maximum range, to 0 for the linear and logistic schedulers.

### 3.3 Environments

For evaluation we use various environments of increasing complexity: Mountain-Car, Inverted-Pendulum-Swingup, Reacher, Hopper, Walker2D, Half-Cheetah. Observation dimensions range from 2 to 26, and action dimensions range from 1 to 6. See Table E.1 for details, including a rough sketch of the reward. The table indicates whether the reward is sparse or dense with respect to a goal state, goal region, or a change of the distance to the goal region. Many environments feature linear or quadratic (energy) penalties on the actions (e.g. Hopper). Penalties on the state can be sparse (such as joint limits), or dense (such as force or required power induced by joint states). Brockman et al. (2016), Coumans & Bai (2016–2021), and Ellenberger (2018) provide further details.

### 3.4 Performed experiments

We evaluated the effects of action noise on the popular and widely-used algorithms: TD3 (Fujimoto et al., 2018), DDPG (Lillicrap et al., 2016), SAC (Haarnoja et al., 2019), and a deterministic version of SAC (DetSAC, Algorithm D.1). Originally SAC was proposed with only exploration from its stochastic policy. However, since SAC is an off-policy algorithm, it is possible to add additional action noise, a common solution for environments such as the Mountain Car. The stochastic policy in SAC typically is a parameterized Gaussian and combining the action noise with the stochasticity of actions sampled from the stochastic policy could impact the results. Thus, we also compared to our DetSAC version, where action noise is added to the mean action of the DetSAC policy (Algorithm D.1).

We used the implementations provided by Raffin et al. (2021a), following the hyper-parameterizations provided by Raffin (2020), but adapting the action noise settings.

The experiments consisted of testing 6 environments, 4 algorithms, 5 noise scales, 3 schedulers and 2 noise types. Each experiment was repeated with 20 different seeds, amounting to 14400 experiments in total. On a single node, `AMD Ryzen 2950X` equipped with four `GeForce RTX 2070 SUPER, 8 GB`, running about twenty experiments in parallel this would amount to a runtime of approximately 244 node-days (which accounted for about 6 weeks on our cluster).

Section I lists further details such as the returns averaged across seeds for each experimental configuration.

### 3.5 Measuring Performance

For each experiment (i.e. single seed), we divided the learning process into 100 segments and evaluated the exploration and learned policy performance for each of those segments. At the end of each segment, we performed evaluation rollouts for 100 episodes or 10000 steps, whichever was reached first, using only complete episodes. This ensures sufficient data points when the episode length varies greatly (e.g. for the Hopper). This procedure was performed for both the deterministic *exploitation* policy as well as the *exploratory* (action noise) policy. The two resulting datasets of evaluation rollouts are used to calculate state-space coverages and returns. These evaluation rollouts, both exploring and exploiting, were *not* used for training and thus do *not* change the amount of training data seen between training steps. We took the mean over these 100 measurements to aggregate them into a single value. This is equivalent to measuring the area under the learning curves. For the evaluation returns, this is called the Performance $P$ and is our main measure for learning performance. Similarly, aggregated evaluation returns measured in this fashion are denoted by $R$.

The learning algorithm uses a noisy (exploratory) policy to collect data and exploratory return and state-space coverage could be assessed based on the replay buffer data. However, to get statistically more robust estimates of the quality of the exploratory policy (returns and state-space coverage), we performed the above mentioned exploratory evaluation rollouts and used these rollouts for assessing state-space coverage

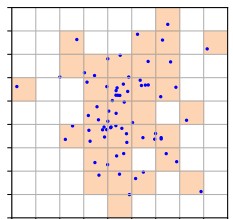
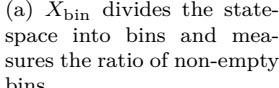
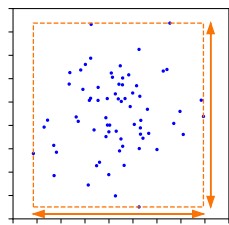
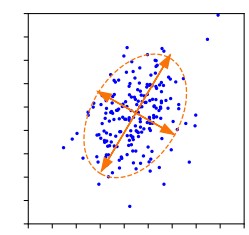
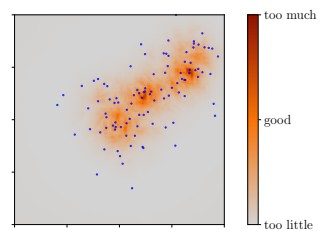

(a) $X_{\text{bin}}$ divides the state-space into bins and measures the ratio of non-empty bins.

(b) $X_{\text{BBM}}$ measures the spread by the mean of the side-lengths of the bounding box.

(c) $X_{\text{NN}}$ measures the spread of the data by the sum of the eigenvalues of the covariance of the data.

(d) $X_{\mathcal{U}\text{rel}}$ (ours) measures the symmetric KL-divergence between a prior over the state space and the collected state-space data.

Figure 3: Illustrations of the state-space coverage measures. $X_{\mathcal{U}\text{rel}}$ scales to high dimensions (unlike $X_{\text{bin}}$) and is not susceptible to estimation artifacts due to points close to the support boundary (unlike $X_{\text{BBM}}$, $X_{\text{NN}}$ and kNN based estimators).

and exploratory returns instead of the data in the replay buffer. Again, these 100 measurements were aggregated by taking the mean and denoted as the exploratory state-space coverage $X$ and the evaluation state-space coverage $E$.

## 3.6 State-Space Coverage

We assess exploration in terms of state-space coverage. We assume that the environment states $s \in \mathcal{R}^d$ have finite upper and lower limits: $\text{low} \le s \le \text{high}$, $\text{low}, \text{high} \in \mathcal{R}^d$. We investigate four measures: $X_{\text{bin}}, X_{\mathcal{U}\text{rel}}, X_{\text{BBM}}, X_{\text{NN}}$, which are illustrated in Figure 3.

The most intuitive measure for state-space coverage is a histogram-based approach $X_{\text{bin}}$, which divides the state space into equally many bins along each dimension and measures the ratio of non-empty bins to the total number of bins:

$$X_{\text{bin}} = \frac{\# \text{ of non-empty bins}}{\# \text{ number of bins}} \tag{5}$$

The number of bins, as the product of divisions along each dimension, grows exponentially with the dimensionality. This means that either the number of bins has to be chosen very low, or, if there are more bins than data points, the ratio has to be adjusted. We chose to limit the number of bins. For a sample of size $m$ and dimensionality $d$ the divisions $k$ along each dimension are chosen to allow for at least $c$ points per bin

$$k = \lfloor * \rfloor \left( \frac{m}{c} \right)^{\frac{1}{d}} \tag{6}$$

However, for high-dimensional data, the number of bins becomes very small and the measure easily reaches 100% and becomes meaningless, or, the required number of data points becomes prohibitively large very quickly. Thus, alternatives are necessary.

Zhan et al. (2019) proposed two state-space coverage measures that also work well in high-dimensional spaces: the *bounding box mean* $X_{\text{BBM}}$, and the *nuclear norm* $X_{\text{NN}}$. $X_{\text{BBM}}$ measures the spread of the data by a $d$ dimensional bounding box around the collected data $D = \{\ldots, \boldsymbol{s}^{(j)}, \ldots\}$ and measuring the mean of the side lengths of this bounding box:

$$X_{\text{BBM}} = \frac{1}{d} \sum_i^d \left[ \max_j s_i^{(j)} - \min_j s_i^{(j)} \right] \tag{7}$$

$X_{\text{NN}}$, the nuclear norm estimates the covariance matrix $C$ of the data and measures data spread by the trace, the sum of the eigenvalues of the estimated covariance:

$$X_{\text{NN}}(D) := \text{trace}\left( C(D) \right) \tag{8}$$

As shown below in Section 3.6.1, extreme values or values close to the state-space boundaries can lead to over-estimation of the state-space coverage by these two measures. We therefore propose a measure more closely related to $X_{\text{bin}}$ but more suitable to higher dimensions: $X_{\mathcal{U}\text{rel}}(D)$. The Uniform-relative-entropy measure $X_{\mathcal{U}\text{rel}}$ assesses the uniformity of the collected data, by measuring the state-space coverage as the symmetric divergence between a uniform prior over the state space $U$ and the data distribution $Q_D$:

$$X_{\mathcal{U}\text{rel}}(D) = -D_{\text{KL}}\big(U||Q_D\big) - D_{\text{KL}}\big(Q_D||U\big) \tag{9}$$

The inspiration for this measure comes from the observation that the exploration reward for count-based methods without task reward would be maximized by a uniform distribution. We assume that for robotics tasks reasonable bounds on the state space can be found. In a bounded state space, the uniform distribution is the least presumptive (maximum-entropy) distribution. The addition of the $D_{\text{KL}}\big(U||Q_D\big)$ term helps to reduce under-estimation of the divergence in areas with low density in $Q_D$. Note that $Q_D$ is only available through estimation, and the support for $Q_D$ is never zero as the density estimate never goes to zero. To estimate the relative uniform entropy we evaluated two divergence estimators, a kNN-based (k-Nearest-Neighbor) estimator and a Nearest-Neighbor-Ratio (NNR) estimator (Noshad et al., 2017). Density estimators based on kNN are susceptible to over- / under-estimation artifacts close to the boundaries (support) of the state space (see Figure B.1 for an illustration). In contrast, the NNR estimator does not suffer from these artifacts. If not specified explicitly, $X_{\mathcal{U}\text{rel}}$ refers to the NNR-based variant.

*kNN $X_{\mathcal{U}rel}$ estimator:* $X_{\mathcal{U}\text{rel}}$ can be estimated using a *kNN density estimate* $\hat{q}_k(s)$, as described in (Bishop, 2006), where $V_d$ denotes the unit volume of a $d$-dimensional sphere, $R_k(x)$ is the Euclidean distance to the $k$-th neighbor of $x$, and $n$ is the total number of samples in $\mathcal{D}$:

$$V_d = \frac{\pi^{d/2}}{\Gamma(\frac{d}{2}+1)} \tag{10}$$

$$\hat{q}_k(x) = \frac{k}{n}\frac{1}{V_d R_k(x)^d} = \frac{k}{nV_d}\frac{1}{R_k(x)^d} \tag{11}$$

where $\Gamma$ denotes the gamma function.

*NNR $X_{\mathcal{U}rel}$ estimator:* Alternatively, $X_{\mathcal{U}\text{rel}}$ can be *estimated using NNR*, an $f$-divergence estimator, based on the ratio of the nearest neighbors around a query point.

For the general case of estimating $D_{\text{KL}}\big(P||Q\big)$, we take samples from $X \sim Q$ and $Y \sim P$. Let $\mathcal{R}_k(Y_i)$ denote the set of the $k$-nearest neighbors of $Y_i$ in the set $Z := X \cup Y$. $N_i$ is the number of points from $X \cap \mathcal{R}_k(Y_i)$, $M_i$ is the number of points from $Y \cap \mathcal{R}_k(Y_i)$, $M$ is the number of points in $Y$ and $N$ is the number of points in $X$, $\eta = \frac{M}{N}$. The NNR measure requires the density of $P$ and $Q$ to be bounded with the lower limit $C_L > 0$, and measures the ratio of points from two different distributions around a query point. Assuming all points of a sample of size $n$ are concentrated around a single point, we lower-bound the density to $C_L = \frac{1}{n}$. To limit the peaks around a single point we upper-bound the densities to $C_U = \frac{n}{1}$.

$$D_{\text{KL}}(P||Q) \approx \hat{D}_g(X,Y) \tag{12}$$

$$\hat{D}_g(X,Y) := \max\left(\frac{1}{M}\sum_{i=1}^{M}\hat{g}\left(\frac{\eta N_i}{M_i+1}\right), 0\right) \tag{13}$$

$$\text{where } \hat{g}(x) := \max\big(g(x), g(C_L/C_U)\big) \tag{14}$$

$$g(\rho) := -\log\rho \tag{15}$$

### 3.6.1 Evaluation of Measures on Synthetic Data

To compare the different exploration measures, we assumed a $d = 25$ dimensional state space, generated data from two different types of distributions, and compared the exploration measures on these data. The experiments were repeated 20 times, and the mean and min-max values are plotted in Figure 4. Each

sampled dataset consists of 5000 points. For most measures the variance is surprisingly small. While the data are $d$-dimensional, they come from factorial distributions, similarly distributed along each dimension. Thus, we can gain intuition about the distribution from scatter plots of the first vs. second dimension. This is depicted at the top of each of the two parts. The bottom part of each comparison shows the different exploration measures, where the scale parameter is depicted on the $x$ axis and the exploration measure on the $y$ axis.

**(a) Growing Uniform:** Figure 4(a) depicts data generated by a uniform distribution, centered around the middle of the state space, with minimal and maximal values growing relatively to the full state space

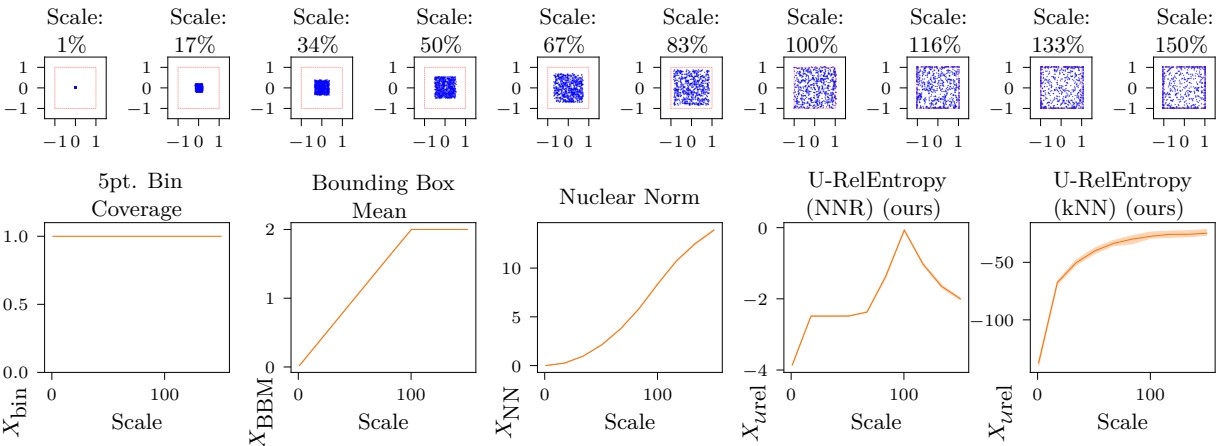

(a) Growing Uniform distribution: evaluation of the state-space coverage measures on synthetic data – for larger scale values more points are clipped to the state-space boundaries, leading to an expected decrease in state-space coverage for scales larger than 100%. This behavior is only captured by $X_{\mathcal{U}\mathrm{rel}}$ (NNR).

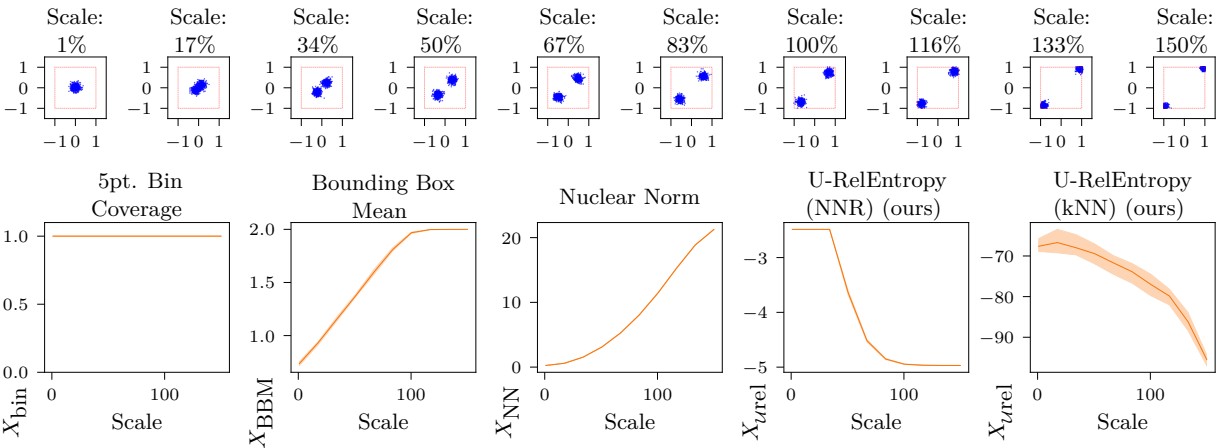

(b) Growing Distance of Modes of 2-Mixture of Truncated Normal: evaluation of the state-space coverage measures on synthetic data. For larger scale values, the location of the mixture components is closer to the boundary – leading to an expected reduction in coverage for larger scale values. $X_{\mathrm{bin}}, X_{\mathrm{BBM}}, X_{\mathrm{NN}}$ fail to capture this behavior.

Figure 4: state-space coverage measures may not accurately represent the real coverage. Each comparison (a-b) shows the different exploration measures $X_{\mathrm{bin}}, X_{\mathrm{BBM}}, X_{\mathrm{NN}}$ and $X_{\mathcal{U}\mathrm{rel}}$ (ours) on synthetic 25 dimensional data. $X_{\mathrm{bin}}$ becomes constant and $X_{\mathrm{BBM}}$ and $X_{\mathrm{NN}}$ suffer from estimation artifacts for points close to the support boundary. The different data generating distributions depend on a *scale* parameter. The distributions are factorial and similarly distributed along each dimension. The scatter plots in (a-b) depict first vs. second dimension. The mean and min-max variation is shown. In the majority of cases the variance is surprisingly small.

according to the *scale* parameter from 1% to 150%. Since in the latter case, many points would lie outside the allowed state space; these values are clipped to the state-space boundaries. This loosely corresponds to an undirectedly exploring agent that overshoots and hits the state-space limits, sliding along the state-space boundaries. Note how the estimation (kNN vs. NNR) has a great impact on the $X_{\mathcal{U}\mathrm{rel}}$ measure's performance here: We would expect a maximum around a scale of 100% and smaller values before and after (due to clipping). Here the $X_{\mathcal{U}\mathrm{rel}}$ (NNR) measure most closely follows this expectation. The ground-truth value of the divergence would follow a similar shape. However, since the densities are limited for the NNR estimator, the ground-truth divergence would show more extreme values.

**(b) Bi-Modal Truncated Normal moving locations:** Figure 4(b) shows a mixture of two truncated Gaussian distributions, with equal standard deviations but located further and further apart (depending on the scale parameter). In this case, the state-space coverage should increase until both distributions are sufficiently far apart, should then stay the same, and finally begin to drop because the proximity to the state-space-boundary limits the points to an ever smaller volume. The inspiration for this example distribution is an agent setting off in two opposite directions and getting stuck at these two opposing limits. While somewhat contrived and more extreme than the inspiring example, it highlights difficulties in the exploration measures. Both the bounding-box mean $X_{\mathrm{BBM}}$ and the nuclear norm $X_{\mathrm{NN}}$ completely fail to account for vastly unexplored areas between the extreme points.

Since the $X_{\mathcal{U}\mathrm{rel}}$ NNR measure is clipped (by definition of NNR) the measure reaches its limits when the density ratios become extreme, which presumably happens for very small and large scale parameters in this setting. The $X_{\mathrm{Urel}}$ kNN approximator is better able to capture the extreme divergence values, however, as pointed out before, this comes at the cost of under-estimating the divergence for points close to the support boundary.

The experiments on synthetic data showed that the histogram based measure is not useful in high-dimensional spaces. The alternatives $X_{\mathrm{BBM}}$ and $X_{\mathrm{NN}}$ are susceptible to artifacts on bounded support. This susceptibility to boundary artifacts is also present in the kNN-based $X_{\mathcal{U}\mathrm{rel}}$ estimator, because of these results we employ the NNR-estimator based $X_{\mathcal{U}\mathrm{rel}}$ in the rest of this paper and refer to it as $X_{\mathcal{U}\mathrm{rel}}$.

## 4 Results: What action noise to use?

In this section we analyze the data collected in the experiments described in Section 3.4. We first look at the experiments performed under a constant scale scheduler since this is the most common case in the literature. In this setting we will look at two aspects: first, is one of the two action noise types generally superior to the other *(Q1)*? And secondly, is there a generally preferable action noise scale *(Q2)*? Then, we compare across constant, linear and logistic schedulers to see if reducing the noise impact over the training process is a reasonable thing to do *(Q3)*. Finally we compare the relative importance of the scheduler, noise type and scale *(Q4)*. See Section F.1 for a brief description of the statistical methods used in this paper and the verification of their assumptions.

### 4.1 (Q1) Which action noise type to use? (and what are the impacts)

To compare the impact of the action noise *type*, we look at the constant $\beta = 1$ case, group the aggregated performance and exploration results (see Section 3.5) by the factors algorithm, environment, and action noise scale and standardize the results to control for their influence. These standardized results are then combined for each noise type. Figure 5 illustrates the results. The comparisons are performed by Welch-t-test, symmetric p-values are listed.

Figure 5 (c) shows that *Ornstein-Uhlenbeck noise leads to increased state-space coverage* under the exploratory policy $X$ as measured by $X_{\mathcal{U}\mathrm{rel}}$. For completeness Figure 5 (d) shows the state-space coverage of the evaluation policy. Here Ornstein-Uhlenbeck increases coverage which might indicate slightly longer trajectories for policies trained under Ornstein-Uhlenbeck noise, however whether this is preferable or not is task dependant. Exploration likely incurs additional costs, e.g. through action penalties, but also by moving the agent away from high-reward-trajectories. Since Ornstein-Uhlenbeck noise is temporally correlated, it

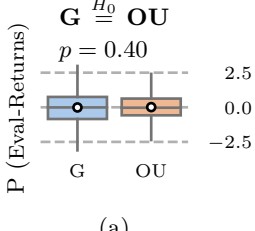
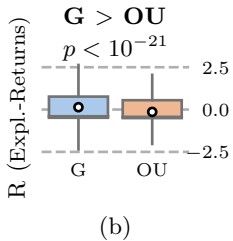
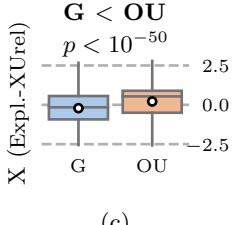
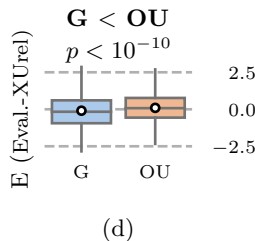

(a)           (b)           (c)           (d)

Figure 5: Comparison of standardized measures ($P$, $R$, $X$, $E$), for Gaussian (G) and Ornstein-Uhlenbeck (OU) noise types, (a-d). Values are standardized to control for and combine algorithm, environment and noise scale: *(a)* For learned performance $P$, measured by evaluation returns, neither of the two noise types is significantly better. *(b)* For Returns collected under the exploration policy $R$, Gaussian noise collects data with slightly better returns ($p < 10^{-21}$). (c) For State-space coverage of the exploratory policy $X$ Ornstein-Uhlenbeck performs better. (d) The State-space coverage of evaluation rollouts $E$ is slightly larger for Ornstein-Uhlenbeck noise without significantly affecting the evaluation returns $P$. Overall neither of the two noise types is superior.

| Environment | P | $p_P$ | $d_{\mathrm{P}}$ | R | $p_R$ | $d_{\mathrm{R}}$ | X | $p_X$ | $d_{\mathrm{X}}$ | E | $p_E$ | $d_{\mathrm{E}}$ |
|---|---|---|---|---|---|---|---|---|---|---|---|---|
| Half-Cheetah | - | 0.89 | - | G | 0.002 | 0.22 | OU | 0.004 | 0.21 | - | 0.20 | - |
| Hopper | OU | $<10^{-3}$ | 0.27 | G | $<10^{-4}$ | 0.29 | G | $<10^{-8}$ | 0.41 | - | 0.71 | - |
| Inverted-Pendulum-Swingup | - | 0.38 | - | G | $<10^{-51}$ | 1.15 | OU | $<10^{-56}$ | 1.22 | G | 0.002 | 0.22 |
| Mountain-Car | OU | $<10^{-10}$ | 0.47 | OU | $<10^{-19}$ | 0.66 | OU | $<10^{-5}$ | 0.34 | OU | $<10^{-21}$ | 0.71 |
| Reacher | G | $<10^{-30}$ | 0.87 | G | $<10^{-26}$ | 0.80 | OU | $<10^{-40}$ | 1.01 | OU | $<10^{-29}$ | 0.84 |
| Walker2D | - | 0.039 | - | G | 0.010 | 0.18 | OU | $<10^{-9}$ | 0.46 | - | 0.28 | - |

Table 2: Per environment the noise type is important: Comparison of Evaluation Returns $P$, Exploratory Returns $R$, Exploratory-$X_{\mathcal{U}\mathrm{rel}}$ $X$, and Evaluation-$X_{\mathcal{U}\mathrm{rel}}$ $E$. Values are standardized to control for and aggregate over algorithm, and noise scale. The results are compared using a Welch-t-test. Significantly better noise type for each environment and measure is reported ($p < 0.01$), as well as two-tailed p-values $p_{(\cdot)}$ and Cohen-d effect size $d_{(\cdot)}$. While overall neither of the two noise types leads to significantly better performance $P$ (see Figure 5), per environment noise type difference is significant.

is more efficient in covering more state-space but also in moving the agent away from high-reward trajectories. Thus *exploratory returns $R$ are larger for Gaussian noise* and conversely smaller for Ornstein-Uhlenbeck noise, see Figure 5 (b). The learning process is able to offset some differences in the data as shown in Figure 5 (a): the significant differences in exploratory returns $R$ and exploratory state-space coverage $X$ *do not* translate into significantly-different performance *across environments*. When viewed on a per-environment basis, Table 2 (column P) shows that, *the preferable noise type depends on the environment*: Ornstein-Uhlenbeck is preferable for Hopper and Mountain-Car, but Gaussian for the Reacher environment. Table 2 (column X) shows that Ornstein-Uhlenbeck leads to larger state-space coverage, as before, and Gaussian noise leads to larger exploratory returns (column R). The only exceptions to this are the Hopper environment, where the Ornstein-Uhlenbeck is more likely to topple the agent and the Mountain-Car environment, where the returns are very closely related to increasing the state-space coverage and thus exhibits an improvement of $R$ by Ornstein-Uhlenbeck noise.

These results show that the *noise type is important* and significantly impacts the performance for *some environments*. Neither of the two noise types leads to better performance, evaluation return $P$, *in general*. However *Ornstein-Uhlenbeck* generally *increases state-space coverage*. This is likely due to the effect, that in many cases the environment acts as an integrator over the actions: in many environments the action constitutes some type of velocity or force control, which by stepping forward, and thus integrating forward in time, amounts to changes in position, or respectively changes in velocity.

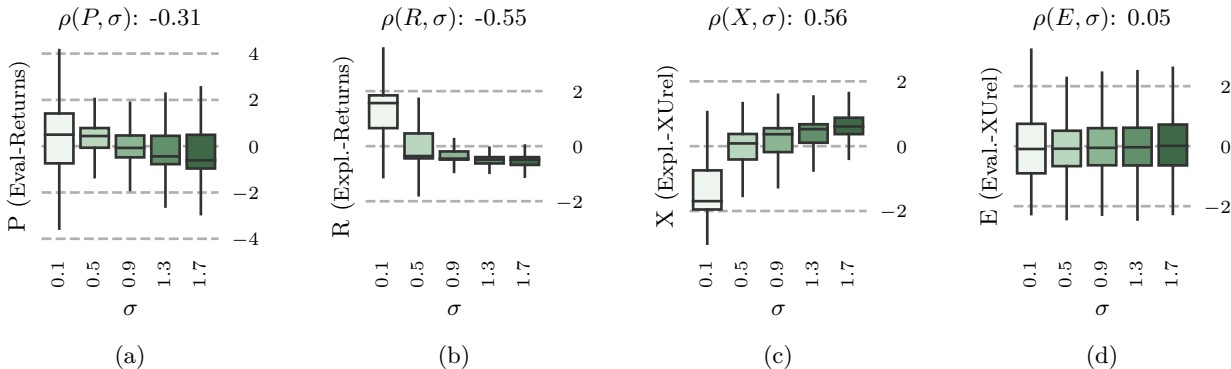

Figure 6: Across environments larger noise scales $\sigma$ are effective in increasing state-space coverage (c), but reduce exploratory returns (b). Measures ($P$, $X$, $R$, $E$) are standardized to control for and aggregate over algorithm, environment and noise type. (a) Evaluation Performance $P$ is negatively correlated with action noise scale ($\rho = -0.31$). (b) Larger noise scales correlate with smaller exploratory returns $R$. (c) Increasing the noise scale $\sigma$ increases exploratory state space coverage $X$. (d) State-space coverage of evaluation rollouts $E$: the learned trajectories appear unaffected by larger noise scale.

| Environment | $\rho(P,R)$ | $\rho(P,X)$ | $\rho(P,\sigma_{\text{scale}})$ | $\rho(R,X)$ | $\rho(R,\sigma_{\text{scale}})$ | $\rho(X,\sigma_{\text{scale}})$ |
|---|---|---|---|---|---|---|
| All | 0.57 | -0.03 | -0.31 | -0.30 | -0.55 | 0.56 |
| Half-Cheetah | 0.22 | -0.28 | -0.35 | -0.64 | -0.74 | 0.75 |
| Hopper | 0.69 | 0.15 | -0.87 | 0.27 | -0.74 | -0.17 |
| Inverted-Pendulum-Swingup | -0.15 | 0.23 | 0.27 | -0.88 | -0.83 | 0.77 |
| Mountain-Car | 0.94 | 0.87 | 0.58 | 0.76 | 0.37 | 0.75 |
| Reacher | 0.84 | -0.88 | -0.56 | -0.96 | -0.84 | 0.69 |
| Walker2D | 0.76 | -0.44 | -0.81 | -0.52 | -0.82 | 0.63 |

Table 3: Data quality, measured by exploratory returns $R$, does not completely determine performance, measured by evaluation returns $P$. $\rho$ denotes Spearman correlation coefficients. Generally $R$ is positively, but surprisingly not always strongly, correlated with $P$. For some environments, exploratory state-space coverage $X$ is beneficial, while generally it is associated with decreased evaluation performance $P$. Across environments and noise types, increasing the noise scale increases exploratory state-space coverage $X$ but reduces exploratory returns $R$.

## 4.2 (Q2) Which action noise scale to use?

To analyze the impact of action noise scale, we look at the constant ($\beta = 1$) case, and control for the impact of the factors algorithm, environment and noise type: by grouping the results according to these factors and standardizing the results. Then results for the same noise scale are combined.

An interesting observation shown in Figure 6 (c) is that state-space coverage of the exploratory policy $X$ correlates positively with action noise scale $\sigma$ ($\rho$ Spearman correlation coefficients). The takeaway from this is that instead of changing the noise type, one might *increase state-space coverage by increasing $\sigma$*. This however leads to a reduction in the exploratory returns $R$, see Figure 6 (c), ($\rho(R,\sigma) = -0.55$). Subsequently, larger noise scales $\sigma$ are associated with decreased learned performance, i.e. smaller evaluation returns $P$, Figure 6 (a), when viewed across environments. Note that for very small noises ($\sigma = 0.1$) the variance of the results $P$ becomes very large. It appears that, in many cases, less noise is actually better, but too little noise often does not work well. A good default for $\sigma$ appears to be $> 0.1$ but $< 0.9$. The scale $\sigma$ does not appear to have a strong effect on the evaluation state-space coverage $E$, Figure 6 (d). When viewed separately for each environment (Table 3), the association between $X$ and $\sigma$ is consistent. The only exception is the Hopper task, where a large noise is more likely to topple the agent, making it fail earlier, thereby reducing state-space coverage. The association between $\rho(R,\sigma)$ is consistently negative, with the exception of the Mountain-Car where more state-space coverage directly translates to higher returns, because the

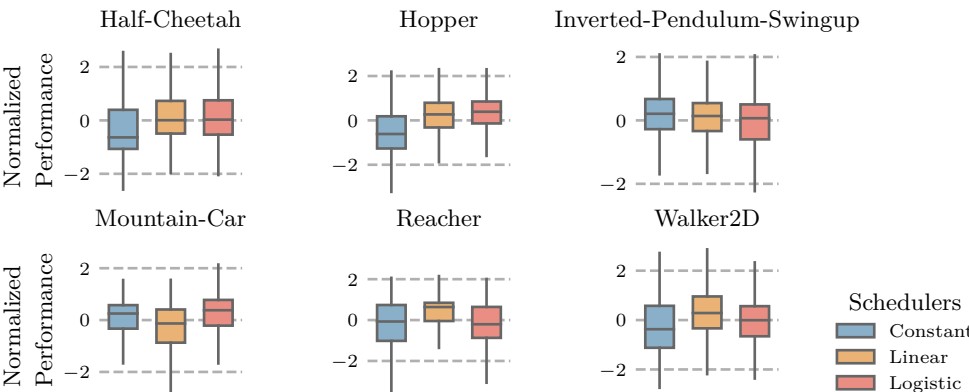

Figure 7: In the majority of cases action noise schedulers improve performance. The figure shows the comparison of the learned policy performance, measured by evaluation returns $P$, for each environment and scheduler. Data is standardized to control for influence of algorithm, environment, noise scale $\sigma$ and noise type. In the majority of cases the linear and logistic schedulers perform better than or comparably to the constant scheduler.

| Scheduler | var($P$) | | | $P$ | | |
| --- | --- | --- | --- | --- | --- | --- |
| | < Constant | < Linear | < Logistic | > Constant | > Linear | > Logistic |
| Constant | 0 | 0 | 0 | 0 | 1 | 1 |
| Linear | 4 | 0 | 1 | 4 | 0 | 2 |
| Logistic | 4 | 1 | 0 | 4 | 1 | 0 |

Table 4: In the majority of cases, using a scheduler reduces variance of the performance (evaluation returns) var($P$), and improves expected performance $P$. The evaluation returns $P$ are standardized to control for the influence of algorithm, noise scale $\sigma$ and noise type. Levene's tests are used to assess difference in variance var($P$) and a multiple-comparison Games-Howell test indicates superior performance $P$. The table shows the number of environments on which each scheduler (row) is significantly better than the other schedulers (column). See Table H.1 for full per-environment results.

environment is underactuated and energy needs to be injected into the system. Offline-RL findings indicate that it is easier to learn from expert data than from data of mixed-quality (Fu et al., 2020). As such, we would expect a very strong correlation between exploratory returns $R$ as a measure of data quality and evaluation returns $P$ as a measure of learned performance. Indeed, $\rho(P, R)$ shows that overall exploratory returns $R$ and evaluation returns $P$ are mostly positively correlated. However, the correlation is not always very strong and can even be negative. This is interesting, because this means that *exploratory returns are not the only determining factor* for learned performance. For example, in the Inverted-Pendulum-Swingup, $\rho(P, R)$ is slightly negative while $\rho(P, X)$ is positive. The results indicate that, the noise scale $\sigma$ has to be chosen to achieve a trade-off between either increasing state-space coverage $X$ or returns $R$ as required for each specific environment.

### 4.3 (Q3) Should we scale down the noise over the training process?

The previous sections indicated that there is no unique solution for the best noise type and that this choice is dependent on the environment. The analysis of the noise scale showed an overall preference for smaller noise scales, but also showed that, in contrast, some environments require more noise to be solved successfully. In this section we analyze schedulers that reduce the influence of action noise ($\beta$) over the training progress.

Figure 7 shows the performance for each environment and each scheduler. The data is normalized by environment and algorithm before aggregation. The general tendency observed across environments is that, when the environment reacts negatively to larger action noise scale (Half-Cheetah, Hopper, Reacher, Walker2D; as shown in Table 3), *reducing the noise impact $\beta$ over time consistently improves performance*. The re-

| Envname | Spearman Correlation | | | $\eta^2$ Effect Size | | |
| --- | --- | --- | --- | --- | --- | --- |
| | $\rho(P, X)$ | $\rho(P, \sigma)$ | $\rho(P, R)$ | $\eta^2_{\text{Scheduler}}$ | $\eta^2_{\text{Type}}$ | $\eta^2_\sigma$ |
| All | -0.503 | -0.120 | 0.770 | 0.005 | 0.000 | **0.084** |
| Mountain-Car | 0.662 | 0.442 | 0.959 | 0.032 | 0.060 | **0.261** |
| Inverted-Pendulum-Swingup | -0.003 | 0.123 | 0.163 | 0.005 | 0.009 | **0.115** |
| Reacher | -0.872 | -0.382 | 0.803 | 0.048 | **0.181** | **0.181** |
| Hopper | -0.349 | -0.599 | 0.651 | 0.045 | 0.022 | **0.660** |
| Walker2D | -0.658 | -0.494 | 0.677 | 0.014 | 0.017 | **0.607** |
| Half-Cheetah | -0.581 | -0.259 | 0.745 | 0.007 | 0.002 | **0.148** |

Table 5: Spearman correlation coefficients and ANOVA $\eta^2$ effect sizes on $P$ for: scheduler, noise type and noise scale $\sigma$. Action noise scale $\sigma$ is associated with the largest effect size for evaluation returns $P$. Results are shown across all environments (standardized and controlled for environment and algorithm, first row), and per environment (standardized and controlled for algorithm). Generally, exploratory returns $R$ and evaluation performance $P$ are positively associated, while generally larger state-space coverage $X$ appears to impact performance $P$ negatively.

*verse* effect appears to be *less important*: for environments benefiting from larger noise scales, the constant scheduler does not consistently outperform the linear and logistic schedulers.

Table 4 shows summarized results indicating the number of environments where scheduler (1), indicated by row, is better than scheduler (2), indicated by column, in terms of variance var$(P)$ and mean performance $P$. See Table H.1 for full results on the pairwise comparisons. Performance differences are assessed by a Games-Howell multiple comparisons test, while variance is compared using Levene's test.

The tests underlying Table 4 show that the differences observed in Figure 7 are indeed significant. Furthermore, the schedulers (linear, logistic) reduce variance var$(P)$ compared to the constant case in four out of six cases. Keeping the impact $\beta$ constant has no beneficial effect on variance in any environment. This indicates that *using a scheduler* to reduce action noise impact increases *consistency in terms of learned performance.*

### 4.4 (Q4) How important are the different parameters?

In the previous sections we looked at each noise configuration parameter independently, first for the constant $\beta$ case (Q1, Q2), secondly for scheduled reduction of $\beta$ (Q3). However, the question remains whether all the parameters are equally important. We standardize results to control for environment and algorithm, and compare across all noise types, noise scales $\sigma$ and all three schedulers.

Table 5 shows Spearman correlation coefficients $\rho(P, X), \rho(P, \sigma), \rho(P, R)$ across all three schedulers (compare to Table 3 which showed correlations for the constant $\beta = 1$ case only). Across environments the schedulers *reduce* correlation $\rho(P, \sigma)$ between learned performance (measured by evaluation returns $P$) and noise scale $\sigma$: from $\rho(P, \sigma) = -0.31$ in the constant scheduler case to $\rho(P, \sigma) = -0.12$ when compared across all three types of schedulers. This is a further indication that using a scheduler increases robustness to $\sigma$. The correlations between $\rho(P, R)$ are increased to 0.77 vs. 0.57, presumably because reducing $\beta$ makes the exploratory policy *more on-policy* and thus $P$ and $R$ become more similar. Interestingly, the schedulers also increase the negative correlation $\rho(P, X)$ between the performance and the exploratory state-space coverage, from $-0.03$ in the constant case to $-0.50$ when viewed across all schedulers. This could be driven by the environments reacting positively to reduced state-space coverage, which under the schedulers achieve more runs high in $R$ but low in $X$, and thus a stronger negative correlation.

The three columns on the right in Table 5 show $\eta^2$ effect sizes of a three-way ANOVA on the evaluation returns $P$: $\eta^2_{\text{Scheduler}}, \eta^2_{\text{Type}}, \eta^2_\sigma$. The $\eta^2$ effect sizes measure the percentage-of-total variance explained by each factor. Only in the Reacher environment, action noise *type* is very important. Surprisingly, in *all cases* the most important factor is *action noise scale*, while the requirement for a large or small action noise scale varies for each environment.

| Envname | Scheduler | $\sigma$ | Type | Horizon | Recommendation |
|---|---|---|---|---|---|
| All | lin | 0.1/0.5 | OU | | |
| Mountain-Car | log | 1.7 | OU | L | large $\sigma$, OU, sched |
| Inverted-Pendulum-Swingup | con | 0.5 | Gauss | L | large $\sigma$ |
| Reacher | lin | 0.1 | Gauss | - | small $\sigma$, Gauss, sched |
| Hopper | lin/log | 0.1 | OU | S | small $\sigma$, sched, OU |
| Walker2D | lin | 0.1 | OU | S | small $\sigma$, OU, sched |
| Half-Cheetah | lin/log | 0.5 | Gauss/OU | S | small $\sigma$ |

Table 6: Comparison of best-ranked noise type, scale and scheduler across all environments and for each environment individually. Scheduler, type and scale are investigated separately by standardizing the values to control for environment, algorithm and the other two respective factors. Horizon indicates whether we expect a long (L) or short (S) effective planning horizon. Recommendation indicates action noise configuration choices in order of importance as per Table 5, for options with effect sizes $\eta^2 > 0.01$ (small effect).

## 5  Discussion & Recommendations

The experiments conducted in this paper showed that the action noise does, depending on the environment, have a *significant* impact on the evaluation performance of the learned policy (Q1). Which action noise type is best unfortunately *depends on the environment*. For the action noise scale (Q2), our results have shown that generally a larger noise scale increases state-space coverage. But since for many environments, learning performance is negatively associated with larger state-space coverage, a large noise scale does not generally have a preferable impact. Similarly, very small scales also appear not to have a preferable impact, as they appear to increase variance of the evaluation performance (Figure 6). However, overall, reducing the action noise scaling factor over time (Q3) mostly has positive effects. Finally we also looked at all factors concurrently (Q4) and found that for most environments noise scale is the most important factor.

It is difficult to draw general conclusions from a limited set of environments and extending the evaluation is limited by the prohibitively large computational costs. However, we would like to provide heuristics derived from our observations that may guide the search for the right action noise. Table 6 shows the best-ranking scheduler, scale and type configurations for each, and across environments. The ranking is based on the count of significantly better comparisons (pairwise Games-Howell test on difference, $p \leq 0.01$, positive test statistic). For each of scheduler, type and scale we standardize to control for the other two factors. Intuitively, the locomotion environments require only a short effective planning horizon: the reward in the environments is based on the distance moved and is relevant as soon as the locomotion pattern is repeated; for example a 30-step horizon is enough for similar locomotion benchmarks (Pinneri et al., 2020). In contrast, the Mountain-Car environment only provides informative reward at the end of a successful episode and thus, the planning horizon needs to be long enough to span a complete successful trajectory (e.g. closer to 100 steps). Similarly, the Inverted-Pendulum-Swingup uses a shaped reward that does not account for spurious local optima: to swing up and increase system energy, the distance to the goal has to be increased again. These observations are indicated in the column *Horizon* (Table 6). Finally, the recommendation column interprets the best-ranked results under the observed importance (Q4) reported in Table 5. Given these results, we provide the following intuitions as a starting point for optimizing the action noise parameters (read as: to address this ▷ do that):

**Environment is under-actuated ▷ increase state-space coverage** We found that in the case of the Mountain-Car and the Inverted-Pendulum-Swingup, both of which are underactuated tasks and require a swinging up phase, larger state-space coverages or larger action noise scales appear beneficial (Table 3 and Table 5). Intuitively, under-actuation implies harder-to-reach state-space areas.

**Reward shape is misleading ▷ increase state-space coverage** Actions are penalized in the Mountain Car by an action-energy penalty, which means not performing any action forms a local optimum. In the case of the Inverted-Pendulum-Swingup, the distance to the goal forms a shaped reward. However, when swinging up, increasing the distance to the goal is necessary. Thus, the shaped reward can be *misleading*:

following the reward gradient to *greedily* leads the agent to a spurious local optimum. Optimizing for a spurious local optimum implies not reaching areas of the state space where the actual goal would be found, thus the state-space coverage needs to be increased to find these areas.

**Horizon is short ▷ reduce state-space coverage** The environments Hopper, Reacher, Walker2D model locomotion tasks with repetitive movement sequences. In the Mountain-Car, positive reward is only achieved at the successful end of the episode, where as in the locomotion tasks positive reward is received after each successful cycle of the locomotion pattern. Thus effectively the required planning horizon is shorter compared to tasks such as the Mountain-Car. Consistently with the previous point, if the effective horizon is shorter, the rewards are shaped more efficiently, we see negative correlations with the state-space coverage and the noise scale: if the planning horizon is shorter, the reward can be optimized more greedily, meaning the state-space coverage can be more focused and thus smaller.

**Need more state-space coverage ▷ increase scale** Our analysis showed that, to increase state-space coverage, one way is to increase the scale of the action noise. This leads to a higher probability of taking larger actions. In continuous control domains, actions are typically related to position-, velocity- or torque-control. In position-control, larger actions are directly related to more extreme positions in the state space. In velocity control, larger actions lead to moving away from the initial state more quickly. In torque control, larger torques lead to more energy in the system and larger velocities. Currently most policies in D-RL are either uni-modal stochastic policies, or deterministic policies. In both cases, larger action noise leads to a broader selection of actions and, by the aforementioned mechanism, to a broader state-space coverage. Note that while this is the general effect we observed, it is also possible that a too large action can have a detrimental effect, e.g. the Hopper falling, and the premature end of the episode will lead to a reduction of the state-space coverage.

**Need more state-space coverage ▷ try Ornstein-Uhlenbeck** Depending on the environment dynamics, correlated noise (Ornstein-Uhlenbeck) can increase the state-space coverage: for example, if the environment shows integrative behavior over the actions, temporally uncorrelated noise (Gaussian) leads to more actions that "undo" previous progress and thus less coverage. Thus correlated Ornstein-Uhlenbeck noise helps to increase state-space coverage.

**Need less state-space coverage or on-policy data ▷ reduce scale | use scheduler to decrease $\beta$** If the policy is already sufficiently good, or the reward is shaped well enough, exploration should focus around good trajectories. This can be achieved using a small noise scale $\sigma$. However, if the environment requires more exploration to find a reward signal, it makes to sense to use a larger action noise scale $\sigma$ in the beginning while gradually reducing the impact of the noise (Q3). The collected data then gradually becomes "more on-policy".

**In general ▷ use a scheduler** We found that using schedulers to reduce the impact of action noise over time, decreases variance of the performance, and thus makes the learning more robust, while also generally increasing the evaluation performance overall. Presumably because, once a trajectory to the goal is found, more fine grained exploration around the trajectory is better able to improve performance.

## 6 Conclusion

In this paper we present an extensive empirical study on the impact of action noise configurations. We compared the two most prominent action noise types: Gaussian and Ornstein-Uhlenbeck, different scale parameters $(0.1, 0.5, 0.9, 1.3, 1.7)$, proposed a scheduled reduction of the impact $\beta$ of the action noise over the training progress and proposed the state-space coverage measure $X_{\mathcal{U}_{\mathrm{rel}}}$ to assess the achieved exploration in terms of state-space coverage. We compared DDPG, TD3, SAC, and its deterministic variant detSAC on the benchmarks Mountain-Car, Inverted-Pendulum-Swingup, Reacher, Hopper, Walker2D, and Half-Cheetah.

We found that (Q1) neither of the two noise types (Gaussian, Ornstein-Uhlenbeck) is generally superior across environments, but that the impact of noise type on learned performance can be significant when viewed separately for each environment: the *noise type* needs to be chosen to *fit the environment*. We found that (Q2) increasing action noise scale, across environments, increases state-space coverage but tends to reduce learned performance. Again, whether state-space coverage and performance are positively correlated,

and thus a larger scale is desired, *depends on the environment*. The positive or negative *correlation should guide the selection* of action noise. Reducing the impact ($\beta$) of action noise over training time (Q3), improves performance in the majority of cases and decreases variance in performance and thus *increases robustness* to the action noise *choice*. Surprisingly, we found (Q4) that the *most important factor* appears to be the action noise *scale* $\sigma$: if less state-space coverage is required, the scale can be reduced. More state-space coverage can be achieved by increasing the action noise scale. This approach is successful even for Gaussian noise on the Mountain-Car. We synthesized our results into a set of *heuristics* on how to choose the action noise based on the properties of the environment. Finally we *recommend a scheduled reduction* of the action noise impact factor $\beta$ of over the training progress to improve robustness to the action noise configuration.

### Acknowledgments

We would like to thank Bart Keulen, David Peer, Onno Eberhard, Sebastian Blaes and the TMLR Reviewers for the useful discussion.

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

## Appendices

## A    A motivating example

The action is generated as $\tilde{a}_t \sim \pi_\theta(s_t)$, $a_t = \tilde{a}_t + \varepsilon_{a_t}$, where $\varepsilon_{a_t}$ denotes the action noise. We calibrate the noise scale to achieve similar returns for both noise types. To calibrate the action noise scale, we assume a constant-zero-action policy upon which the action noise is added and effectively use $a_t = \varepsilon_{a_t}$ as the action sequence. We find that a scale of about 0.6 for Gaussian action noise and a scale of about 0.5 for Ornstein-Uhlenbeck noise lead to a mean return of about $-30$. This is shown in Table 1. A successful solution to the Mountain-Car environment yields a positive return $0 < \sum r_t < 100$. We then use these two noise configurations and perform learning with DDPG, SAC and TD3. The resulting learning curves are shown in Figure 1 and very clearly depict the huge impact the noise configuration has: with similar returns of the noise-only policies, we achieve substantially different learning results, either leading to failure or success on the task.

To achieve a swing-up, the actions must not change direction too rapidly but rather need to change direction with the right frequency. Ornstein-Uhlenbeck noise is temporally correlated and thus helps solving the environment successfully with a smaller scale $\sigma$. In this environment, the algorithms tend to converge either to the successful solution of the environment by swinging up, or to a passive zero-action solution which incurs no penalty.

## B    Boundary Artifacts

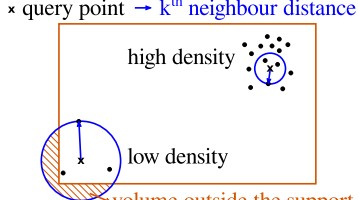

Figure B.1: k-nearest-neighbor density estimators suffer from boundary artifacts when estimating densities with bounded support. The density around a query point is estimated by the volume required to include the $k$ nearest points. Top right, high density region: the volume to include $k$-points is smaller when the density is high. bottem left, low density: in lower densities a larger volume is required to include $k$ points. This also illustrates the boundary artifacts: when querying the density close to the support boundary, part of the query volume is outside the support. Thus the volume required to contain $k$ points is over-estimated. This problem is amplified in higher dimensional spaces as the boundary artifacts occur as soon as any single dimension of the sphere protrudes outside the support.

## C    Action Noise in SAC

SAC as defined by (Haarnoja et al., 2019) does not use action noise for exploration. Instead, actions are sampled from a stochastic Gaussian policy However, since SAC is an off-policy algorithm, additive action noise can additionally be used. The SAC algorithm uses a target entropy parameter. The entropy coefficient of SAC is trained such that the average entropy of the Gaussian policy matches this target. In the implementation we use (Raffin et al., 2021a), the entropy target can be automatically chosen based on the size of the action space. In the Mountain-Car this amounts to a target entropy of 1. The entropy of a Gaussian is defined as $\mathcal{H}_\mathcal{N}(\sigma) = \ln\left(\sigma\sqrt{2\,\pi\,e}\right)$. A $\sigma = 1.7$ approximately translates to an entropy target of 1.95.

In SAC the value function $V$ contains an additional entropy bonus term: $V(s_t) = \mathbb{E}_{a_t \sim \pi}\left[Q(s_t, a_t)\right] + \alpha\mathcal{H}(\pi(\cdot|s_t))$. This term is weighted by the entropy coefficient $\alpha$. Additionally, the SAC policy is defined as a softmax operation over the Q function: $\pi_{\text{softmax}}(a_t|s_t) = \frac{\exp(\frac{1}{\alpha}Q(s_t, a_t))}{Z(s_t)}$ where $Z$ is a normalizing term, chosen s.t. $\int \pi_{\text{softmax}}(a|s_t)\,\mathrm{d}a = 1$. Here, the entropy coefficient $\alpha$ plays a double role, in both the entropy bonus and the softness of the softmax operation. Thus, increasing the scale of the Gaussian has a direct influence on the smoothness of the softmax and can thus change the learning performance. Using action noise is independent of the softmax and can be tuned independently. Furthermore, action noise allows for the use of a correlated noise process, which in the case of the Mountain-Car has a large beneficial influence. This explains why using action noise can be beneficial even for stochastic policies.

| Environment | Algorithm | Type | Scale | Entropy Target | P | X | R | E |
|---|---|---|---|---|---|---|---|---|
| Mountain-Car | SAC | - | | 1.95 | -34 | -2.85 | -34 | -2.86 |
| | | | | auto | -7 | -4.28 | -7 | -4.29 |
| | | Gauss | 0.1 | auto | -5 | -4.15 | -6 | -4.22 |
| | | | 0.5 | auto | 3 | -2.94 | -18 | -3.93 |
| | | | 0.9 | auto | 17 | -2.27 | -18 | -3.55 |
| | | | 1.3 | auto | 23 | -2.06 | -21 | -3.38 |
| | | | 1.7 | auto | 24 | -1.97 | -25 | -3.34 |
| | | OU | 0.1 | auto | -1 | -3.94 | -3 | -4.11 |
| | | | 0.5 | auto | 51 | -1.80 | 37 | -2.62 |
| | | | 0.9 | auto | 68 | -1.49 | 53 | -2.22 |
| | | | 1.3 | auto | 72 | -1.42 | 57 | -2.21 |
| | | | 1.7 | auto | 73 | -1.39 | 57 | -2.14 |

Table C.1: Comparison of SAC with Action noise against SAC relying on the stochastic policy for exploration. Increasing the entropy target increases the state space coverage.

## D  Deterministic SAC

---

**Algorithm D.1** (Deterministic) Soft Actor-Critic

---

Initialize parameter vectors $\psi$, $\bar{\psi}$, $\theta$, $\phi$.
**for** each iteration **do**
    **for** each environment step **do**
        $\mu_t, \sigma_t = f_\phi(s_t)$
        $\varepsilon_t \sim \mathcal{A}$                                   $\triangleright \mathcal{A} \dots$ action noise process
        $a_t = \mu_t + \varepsilon_t$                                      $\triangleright$ DetSAC
        $\pi_\phi(\cdot|s_t) = \mathcal{N}(\cdot|\mu_t, \sigma_t)$                           $\triangleright$ SAC
        $a'_t \sim \pi_\phi(\cdot|s_t)$
        $a_t = a'_t + \varepsilon_t$
        $s_{t+1} \sim p(s_{t+1}|s_t, a_t)$
        $\mathcal{D} \leftarrow \mathcal{D} \cup \big\{(s_t, a_t, r(s_t, a_t), s_{t+1})\big\}$
    **end for**
    **for** each gradient step **do**
        $\dots$ *original SAC update* (Haarnoja et al., 2019)
    **end for**
**end for**

---

# E   Benchmark Environments

| Environment | Illustration | $\dim(\mathcal{O})$ | $\dim(\mathcal{A})$ | Reward | | | |
|---|---|---|---|---|---|---|---|
| Mountain-Car |  | 2 | 1 | $\mathbf{1}(s_t, s_G)$ | $-\lvert a_t \rvert_2^2$ | | |
| Inverted-Pendulum-Swingup |  | 5 | 1 | $\lvert \varphi(s_t) - \varphi_G \rvert_1$ | | | |
| Reacher |  | 9 | 2 | $\nabla^- \lvert s_t - s_G \rvert_2$ | $-\lvert \varphi(s_t) \rvert_2^2$ | $-\mathbf{1}(\varphi(s_t), \varphi_{\text{limit}})$ | $-\lvert a_t \rvert_1$ |
| Hopper |  | 15 | 3 | $\nabla^- \lvert s_t - s_G \rvert_1$ | $-\lvert \varphi(s_t) \rvert_2^2$ | $-\mathbf{1}(\varphi(s_t), \varphi_{\text{limit}})$ | $-\lvert a_t \rvert_1$ |
| Walker2D |  | 22 | 6 | $\nabla^- \lvert s_t - s_G \rvert_1$ | $-\lvert \varphi(s_t) \rvert_2^2$ | $-\mathbf{1}(\varphi(s_t), \varphi_{\text{limit}})$ | $-\lvert a_t \rvert_1$ |
| Half-Cheetah |  | 26 | 6 | $\nabla^- \lvert s_t - s_G \rvert_1$ | $-\lvert \varphi(s_t) \rvert_2^2$ | $-\mathbf{1}(\varphi(s_t), \varphi_{\text{limit}})$ | $-\lvert a_t \rvert_1$ |

Table E.1: Benchmarks environments used in our evaluation in increasing order of complexity. $\lvert \mathcal{O} \rvert$ denotes Observation space dimensions. $\lvert \mathcal{A} \rvert$ denotes Action space dimensions. Explanation of Reward components: $\mathbf{1}(b, c)$ indicator function (sparse reward or penalty) of $b$ w.r.t. to the set $c$; $\lvert b \rvert_n$ n-norm of $b$; $\varphi(s_t)$ angular component of state; $\nabla^- b$ finite-difference reduction of $b$ between time-steps; $\varphi_{\text{max}}$ joint limit; $s_G$ goal state; $\lvert \varphi(s_t) \rvert_2^2$ denotes an angular-power-penalty. Factors in the reward are omitted. Distances e.g. $\lvert s_t - s_G \rvert_n$ may refer to a subspace of the vector $s_t$. Section 3.3

# F  Statistical Methods

| Section | Where | Statistic | $n/N$ | non-Normal | $\neq$ Variance |
|---|---|---|---|---|---|
| Section 4.1 | Figure 5 | Welch t-Test | 2400/4800 | CLT | Robust |
|  | Table 2 | Welch t-Test | 400/800 | CLT | Robust |
| Section G | Table G.1 | Mann-Whitney-U Test | 400/800 | Robust | Robust |
| Section 4.3 | Table 4 | Levene's Test | 800/2400 | Robust | - |
|  | Table 4 | Games-Howell-Test | 800/2400 | $\alpha \cdot 0.2$ | Robust |
| Section 5 | Table 6 (All) Scheduler | Games-Howell-Test | 4800/14400 | $\alpha \cdot 0.2$ | Robust |
| Section 5 | Table 6 (All) Scale | Games-Howell-Test | 2880/14400 | $\alpha \cdot 0.2$ | Robust |
| Section 5 | Table 6 (All) Type | Games-Howell-Test | 3600/14400 | $\alpha \cdot 0.2$ | Robust |
| Section 5 | Table 6 (Env) Scheduler | Games-Howell-Test | 800/2400 | $\alpha \cdot 0.2$ | Robust |
| Section 5 | Table 6 (Env) Scale | Games-Howell-Test | 480/2400 | $\alpha \cdot 0.2$ | Robust |
| Section 5 | Table 6 (Env) Type | Games-Howell-Test | 1200/2400 | $\alpha \cdot 0.2$ | Robust |

Table F.1: Summary of applied tests, per group sample size $n$ and cumulative size across groups $N$, see Section F.1 about the $\alpha$ adjustment in the Games-Howell test. For large sample sizes the t-statistic approaches a normal distribution (CLT). Sample sizes of 30 (Boneau, 1960) are usually assumed to be large enough. (Lumley et al., 2002) provide further evidence for the adequacy of our sample sizes.

## F.1  Statistical Methods Details

We use statistical methods implemented in (Jones et al., 2001; Vallat, 2018) as well as our own implementations.

**Welch t-test** : does not assume equal variance. Reporting two-tailed p-value. Significant for one-tailed when $\frac{p}{2} < \alpha$.

**Games-Howell test** Performing multiple comparisons with a t-test increases the risk of Type I errors. To control for Type I errors, the Games-Howell test (Games & Howell, 1976), a multiple-comparison test applicable to cases with heterogeneity of variance, should be used (Sauder & DeMars, 2019). Sample sizes should be $n \geq 6$ in each group.

The test statistic $t$ is distributed according to Tukey's studentized range $q$. (Games & Howell, 1976) describe that the test has been found to be robust to non-normality by (Ramseyer & Tcheng, 1973), especially in the case of equal sample sizes. This holds in our case. (Ramsey et al., 2011) have found the concurrent violation of homogeneity of variance and non-normality can increase type-I errors. Their results indicate that an error level of $\alpha = 0.05$ can be achieved by applying a reduction of the significance level of $\alpha$ and find controlling for this error by reducing the significance level to $0.38\alpha$. Further evidence for reducing the significance threshold to 0.01 in order to achieve error rates $< 0.05$ is provided by (Ramseyer & Tcheng, 1973).

$$t = \frac{\bar{x}_i - \bar{x}_j}{\sigma} \tag{16}$$

$$\sigma = \sqrt{\left( \frac{s_i^2}{n_i} + \frac{s_j^2}{n_j} \right)} \tag{17}$$

$$df = \frac{\left( \frac{s_i^2}{n_i} + \frac{s_j^2}{n_j} \right)^2}{\frac{\left( \frac{s_i^2}{n_i} \right)^2}{n_i - 1} + \frac{\left( \frac{s_j^2}{n_j} \right)^2}{n_j - 1}} \tag{18}$$

$$\tag{19}$$

The $p$-value is then calculated for $k$ sample-groups as

$$q_{t \cdot \sqrt{2}, k, df} \tag{20}$$

**ANOVA** We perform a *balanced* N-way ANOVA, i.e. with N independent factors, each with multiple levels (categorical values). Since the study design is balanced this is equivalent to a type-I ANOVA in which the order of terms does not matter (because the design is balanced).

**Eta squared $\eta^2$** The effect size eta squared $\eta^2$ denotes the relative variance explained by a factor to the total variance observed: $\eta^2 = \frac{SS_{C(x)}}{SS_{\text{Total}}}$

|           | DF    | Sum of Squares |       F | PR(>F)   |
|-----------|-------|----------------|---------|----------|
| C(y)      | 9.0   | 4167.583       | 478.576 | 0        |
| C(x)      | 9.0   | 91.118         | 10.463  | 1.7e-15  |
| C(y):C(x) | 81.0  | 81.172         | 1.036   | 0.397    |
| Residual  | 901.0 | 871.798        |         |          |
| Total     |       | 5211.672       |         |          |

Table F.2: ANOVA example. The partial $\eta^2$ for a factor is calculated as the sum of squares, variance explained by that factor, divided by the sum of the variance explained plus the unexplained residual variance.

Effect sizes are interpreted as:

$$\eta^2 \geq 0.01 \quad \text{small effect} \tag{21}$$
$$\eta^2 \geq 0.06 \quad \text{medium effect} \tag{22}$$
$$\eta^2 \geq 0.14 \quad \text{large effect} \tag{23}$$
$$\tag{24}$$

**Levene's Test** assesses (un)equality of group variances.

$$z_{ij} = |y_{ij} - \tilde{y}_j| \tag{25}$$
$$F = \frac{N - p}{p - 1} \frac{\sum_{j=1}^{p} n_j (\tilde{z}_j - \tilde{z})^2}{\sum_{j=1}^{p} \sum_{i=1}^{n_j} (z_{ij} - \tilde{z}_j)^2} \tag{26}$$
$$d_1 = p - 1 \tag{27}$$
$$d_2 = N - p \tag{28}$$
$$\tilde{z}_j = \frac{1}{n_j} \sum_{i=1}^{n_j} z_{ij} \tag{29}$$
$$\tilde{z} = \frac{1}{N} \sum_{j=1}^{p} \sum_{i=1}^{n_j} z_{ij} \tag{30}$$

where $p$ is the number of groups, $n_j$ is the size of group $j$ and $N$ is the total number of observations. $\tilde{y}_j$ is the median of group $j$, $z_{ij}$ denotes sample $i$ in group $j$. The $F$ statistic follows the F-distribution with degrees of freedom $d_1, d_2$.

This variant of Levene's test, $\tilde{y}_j$ median instead of mean, is also called Brown-Forsythe test (Brown & Alan B. Forsythe, 1974) and is more robust to non-normal distributions.

**Cohen-d effect size** : Cohen-d is illustrated in Figure F.1 and measures the distance of the means of two sample groups normalized to the pooled variance:

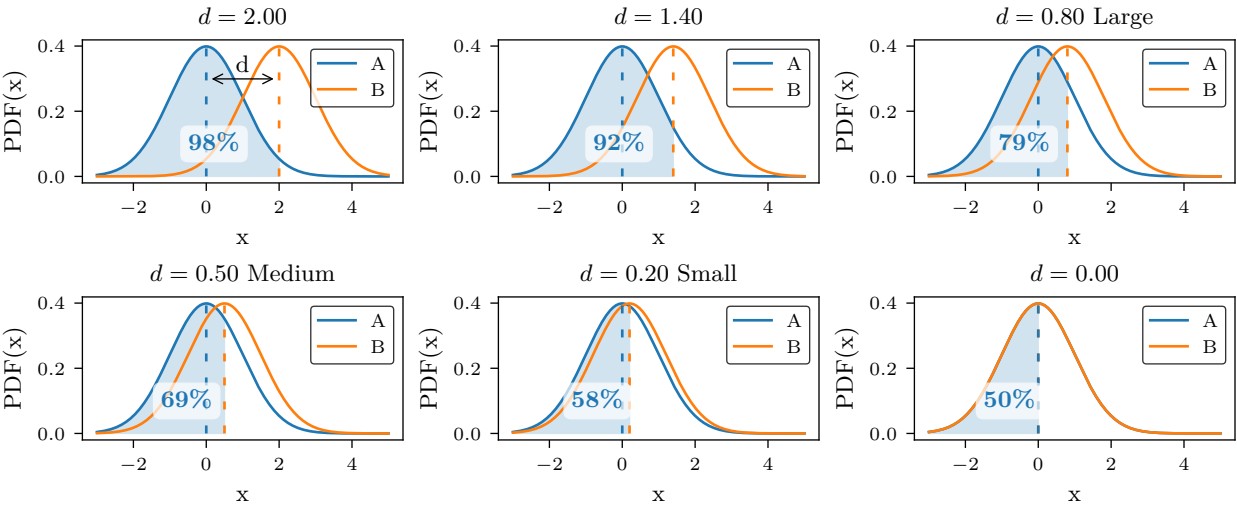

Figure F.1: Illustration of Cohen-d effect size: the Cohen-d measures the standardized difference between the means of two groups, equivalent to a z-score. Effect sizes $d \geq 0.2$ are called small, $d \geq 0.5$ medium, $d \geq 0.8$ large effects. Under equal-variance Gaussian assumption this can be interpreted as $n$-percent of group A below the mean of group B. Illustrated as the shaded area.

$$\text{Effect size} = \frac{[\text{Mean Group A}] - [\text{Mean Group B}]}{\text{Pooled Std Deviation}} \tag{31}$$

$$d = \frac{\bar{x}_1 - \bar{x}_2}{s} \tag{32}$$

$$s = \sqrt{\frac{(n_1 - 1)s_1^2 + (n_2 - 1)s_2^2}{n_1 + n_2 - 2}} \tag{33}$$

## G (Q1) Which action noise type to use? – Mann-Whitney-U Test

| Environment | P | $p_P$ | $d_P$ | R | $p_R$ | $d_R$ | X | $p_X$ | $d_X$ | E | $p_E$ | $d_E$ |
|---|---|---|---|---|---|---|---|---|---|---|---|---|
| Half-Cheetah | - | 0.08 | - | - | 0.86 | - | OU | 0.005 | 0.21 | - | 0.28 | - |
| Hopper | OU | <$10^{-5}$ | 0.27 | G | <$10^{-4}$ | 0.29 | G | <$10^{-10}$ | 0.41 | - | 0.69 | - |
| Inverted-Pendulum-Swingup | - | 0.040 | - | G | <$10^{-40}$ | 1.15 | OU | <$10^{-47}$ | 1.22 | - | 0.32 | - |
| Mountain-Car | OU | <$10^{-10}$ | 0.47 | OU | <$10^{-15}$ | 0.66 | OU | <$10^{-11}$ | 0.34 | OU | <$10^{-21}$ | 0.71 |
| Reacher | G | <$10^{-32}$ | 0.87 | G | <$10^{-21}$ | 0.80 | OU | <$10^{-35}$ | 1.01 | OU | <$10^{-26}$ | 0.84 |
| Walker2D | OU | <$10^{-5}$ | 0.15 | G | 0.006 | 0.18 | OU | <$10^{-13}$ | 0.46 | G | <$10^{-6}$ | 0.08 |

Table G.1: Comparison of noise types as in Table 2, p-values of Mann-Whitney-U test are reported instead of the Welch-t-test. Similar tendencies are shown.

# H    Impact of Scheduler on Variance and Learned Performance

| Scheduler | Envname | var($P$) < Constant | < Linear | < Logistic | $P$ > Constant | > Linear | > Logistic |
|---|---|---|---|---|---|---|---|
| Constant | Half-Cheetah | | No | No | | No | No |
| | Hopper | | No | No | | No | No |
| | Inverted-Pendulum-Swingup | | No | No | | No | Yes $p < 10^{-5}$ |
| | Mountain-Car | | No | No | | Yes $p < 10^{-6}$ | No |
| | Reacher | | No | No | | No | No |
| | Walker2D | | No | No | | No | No |
| Linear | Half-Cheetah | Yes $p < 10^{-3}$ | | No | Yes $p < 10^{-6}$ | | No |
| | Hopper | Yes $p < 10^{-6}$ | | No | Yes $p < 10^{-6}$ | | No |
| | Inverted-Pendulum-Swingup | No | | No | No | | No |
| | Mountain-Car | No | | No | No | | No |
| | Reacher | Yes $p < 10^{-29}$ | | Yes $p < 10^{-15}$ | Yes $p < 10^{-6}$ | | Yes $p < 10^{-6}$ |
| | Walker2D | Yes $p < 10^{-4}$ | | No | Yes $p < 10^{-6}$ | | Yes $p < 10^{-6}$ |
| Logistic | Half-Cheetah | Yes $p < 10^{-4}$ | No | | Yes $p < 10^{-6}$ | No | |
| | Hopper | Yes $p < 10^{-8}$ | No | | Yes $p < 10^{-6}$ | No | |
| | Inverted-Pendulum-Swingup | No | No | | No | No | |
| | Mountain-Car | No | Yes $p = 0.002$ | | Yes $p < 10^{-5}$ | Yes $p < 10^{-6}$ | |
| | Reacher | Yes $p < 10^{-4}$ | No | | No | No | |
| | Walker2D | Yes $p < 10^{-6}$ | No | | Yes $p < 10^{-4}$ | No | |
| Constant | Sum | 0 | 0 | 0 | 0 | 1 | 1 |
| Linear | Sum | 4 | 0 | 1 | 4 | 0 | 2 |
| Logistic | Sum | 4 | 1 | 0 | 4 | 1 | 0 |

Table H.1: In the majority of cases, using a scheduler *reduces variance* var($P$) of the performance (evaluation returns), and *improves* expected performance $P$. The rows shows whether "Scheduler" is significantly better than the scheduler indicated in the columns var($P$) and $P$. The evaluation returns $P$ are standardized to control for the influence of algorithm, noise scale $\sigma$ and noise type. Levene's test is used to assess difference in variance var($P$) and a multiple-comparison Games-Howell test indicates superior performance $P$.

# I    Performed experiments

This section lists the achieved final returns, calculated as the average of the evaluation returns of the last 5 out of our 100 training segments, for each noise setting, for the constant Table I.1, linear Table I.2, and logistic Table I.1 schedulers.

Each noise configuration is repeated with 20 random seeds. Table I.4 lists the performance $P$ for each noise configuration as the mean across the 20 seeds.

Choosing a fixed set of 20 different seeds could introduce randomness artifacts, making one algorithm *appear* to perform better than the other. To combat these biases, each run was performed from an independently, randomly drawn seed. The seeds are sampled using `os.urandom`, which provides a string of random bytes suitable for cryptographic use. This should be sufficient to ensure independence between seeds.

| | | | Return | | | | | | | | | |
|---|---|---|---|---|---|---|---|---|---|---|---|---|
| | | Scale | 0.1 | 0.5 | 0.9 | 1.3 | 1.7 | 0.1 | 0.5 | 0.9 | 1.3 | 1.7 |
| | | Type | Gauss | Gauss | Gauss | Gauss | Gauss | OU | OU | OU | OU | OU |
| Scheduler | Environment | Algorithm | | | | | | | | | | |
| Constant | Half-Cheetah | DDPG | 179 | 269 | 392 | 417 | 341 | 165 | 214 | 279 | 293 | 289 |
| | | DetSAC | 1719 | 1741 | 857 | 587 | 590 | 1619 | 1578 | 1156 | 787 | 731 |
| | | SAC | 1702 | 1856 | 531 | 389 | 292 | 1906 | 1550 | 1139 | 766 | 680 |
| | | TD3 | 1891 | 1651 | 1158 | 856 | 701 | 1582 | 1470 | 975 | 940 | 722 |
| | Hopper | DDPG | 1113 | 541 | 348 | 344 | 276 | 1131 | 808 | 666 | 507 | 461 |
| | | DetSAC | 2170 | 1101 | 837 | 572 | 542 | 2159 | 1602 | 1345 | 966 | 836 |
| | | SAC | 2291 | 979 | 808 | 717 | 604 | 2298 | 1599 | 1330 | 992 | 875 |
| | | TD3 | 2294 | 1833 | 1525 | 1370 | 1186 | 2258 | 1873 | 1539 | 1213 | 1164 |
| | Inverted-Pendulum-Swingup | DDPG | 819 | 819 | 816 | 827 | 818 | 822 | 859 | 879 | 877 | 878 |
| | | DetSAC | 838 | 884 | 880 | 886 | 887 | 842 | 887 | 886 | 888 | 888 |
| | | SAC | 881 | 887 | 883 | 888 | 884 | 886 | 888 | 889 | 887 | 888 |
| | | TD3 | 868 | 883 | 881 | 882 | 883 | 873 | 879 | 882 | 884 | 886 |
| | Mountain-Car | DDPG | -0 | 52 | 84 | 67 | 74 | -0 | 94 | 84 | 26 | 56 |
| | | DetSAC | 5 | 14 | 37 | 47 | 78 | 13 | 85 | 94 | 94 | 94 |
| | | SAC | 4 | 23 | 42 | 60 | 52 | 9 | 92 | 94 | 94 | 94 |
| | | TD3 | 5 | 94 | 65 | 74 | 74 | -0 | 94 | 74 | 74 | 94 |
| | Reacher | DDPG | 17 | 18 | 18 | 18 | 17 | 16 | 15 | 14 | 13 | 13 |
| | | DetSAC | 19 | 19 | 19 | 19 | 19 | 18 | 18 | 17 | 16 | 15 |
| | | SAC | 18 | 18 | 17 | 16 | 16 | 18 | 17 | 16 | 15 | 13 |
| | | TD3 | 17 | 16 | 15 | 15 | 14 | 17 | 16 | 15 | 15 | 14 |
| | Walker2D | DDPG | 787 | 519 | 391 | 263 | 321 | 849 | 759 | 443 | 471 | 465 |
| | | DetSAC | 1814 | 1211 | 424 | 305 | 311 | 1616 | 1139 | 740 | 661 | 646 |
| | | SAC | 1883 | 1233 | 448 | 392 | 383 | 1710 | 1068 | 712 | 691 | 641 |
| | | TD3 | 2001 | 1878 | 1618 | 1321 | 1228 | 1821 | 1870 | 1596 | 1353 | 1230 |

Table I.1: This table shows the final evaluation return (mean over last five percent of training) for each configuration under the Constant regime. The mean across all 20 runs is reported.

| | | | Return | | | | | | | | | |
|---|---|---|---|---|---|---|---|---|---|---|---|---|
| | | Scale | 0.1 | 0.5 | 0.9 | 1.3 | 1.7 | 0.1 | 0.5 | 0.9 | 1.3 | 1.7 |
| | | Type | Gauss | Gauss | Gauss | Gauss | Gauss | OU | OU | OU | OU | OU |
| Scheduler | Environment | Algorithm | | | | | | | | | | |
| Linear | Half-Cheetah | DDPG | 158 | 155 | 154 | 150 | 142 | 176 | 142 | 138 | 127 | 124 |
| | | DetSAC | 1826 | 2538 | 1131 | 888 | 924 | 1547 | 1989 | 2062 | 1564 | 1657 |
| | | SAC | 1991 | 2471 | 1331 | 791 | 860 | 2249 | 2217 | 2062 | 1784 | 1735 |
| | | TD3 | 1957 | 2366 | 2034 | 1658 | 1731 | 1976 | 2069 | 2168 | 1881 | 1339 |
| | Hopper | DDPG | 1039 | 1071 | 883 | 864 | 790 | 965 | 937 | 999 | 964 | 871 |
| | | DetSAC | 2276 | 1923 | 1634 | 1540 | 1318 | 2308 | 1913 | 1702 | 1484 | 1297 |
| | | SAC | 2288 | 1840 | 905 | 814 | 789 | 2208 | 1899 | 1697 | 1579 | 1341 |
| | | TD3 | 2057 | 2107 | 1688 | 1581 | 1610 | 2099 | 2051 | 1943 | 1818 | 1633 |
| | Inverted-Pendulum-Swingup | DDPG | 836 | 830 | 832 | 838 | 835 | 823 | 817 | 829 | 830 | 750 |
| | | DetSAC | 867 | 882 | 885 | 878 | 882 | 778 | 887 | 887 | 888 | 888 |
| | | SAC | 884 | 883 | 879 | 881 | 888 | 879 | 888 | 886 | 888 | 887 |
| | | TD3 | 866 | 857 | 836 | 845 | 830 | 850 | 859 | 866 | 868 | 857 |
| | Mountain-Car | DDPG | -0 | 56 | 46 | 31 | 65 | -0 | 67 | 55 | 26 | 45 |
| | | DetSAC | 13 | -0 | 5 | 28 | 38 | 5 | 33 | 95 | 95 | 95 |
| | | SAC | -1 | 4 | 9 | 37 | 18 | 4 | 71 | 90 | 94 | 95 |
| | | TD3 | -0 | -0 | 37 | 56 | 84 | -0 | 79 | 74 | 74 | 74 |
| | Reacher | DDPG | 16 | 16 | 16 | 16 | 16 | 16 | 16 | 16 | 16 | 15 |
| | | DetSAC | 18 | 19 | 20 | 19 | 20 | 19 | 18 | 17 | 17 | 17 |
| | | SAC | 18 | 18 | 18 | 18 | 18 | 18 | 18 | 17 | 17 | 17 |
| | | TD3 | 17 | 18 | 19 | 19 | 19 | 17 | 17 | 17 | 16 | 17 |
| | Walker2D | DDPG | 791 | 755 | 635 | 476 | 360 | 820 | 839 | 661 | 642 | 460 |
| | | DetSAC | 1602 | 1348 | 1048 | 970 | 894 | 1656 | 1276 | 1167 | 1014 | 876 |
| | | SAC | 1678 | 1630 | 845 | 637 | 652 | 1859 | 1334 | 1164 | 945 | 1028 |
| | | TD3 | 1780 | 1660 | 1584 | 1254 | 1238 | 1820 | 1766 | 1765 | 1647 | 1622 |

Table I.2: This table shows the final evaluation return (mean over last five percent of training) for each configuration under the Linear scheduler regime. The mean across all 20 runs is reported.

| Scheduler | Environment | | Scale | 0.1 | 0.5 | 0.9 | 1.3 | Return 1.7 | 0.1 | 0.5 | 0.9 | 1.3 | 1.7 |
| | | | Type | Gauss | Gauss | Gauss | Gauss | Gauss | OU | OU | OU | OU | OU |
| | | Algorithm | | | | | | | | | | | |
| Logistic | Half-Cheetah | DDPG | | 158 | 156 | 143 | 136 | 124 | 180 | 153 | 129 | 128 | 108 |
| | | DetSAC | | 1639 | 2168 | 1923 | 1559 | 1208 | 1680 | 2100 | 1921 | 1719 | 1235 |
| | | SAC | | 1995 | 2199 | 1984 | 1746 | 1413 | 2158 | 2217 | 1927 | 1613 | 1511 |
| | | TD3 | | 1694 | 2329 | 1864 | 1373 | 1363 | 2030 | 2047 | 1954 | 2002 | 1682 |
| | Hopper | DDPG | | 902 | 1181 | 1103 | 1246 | 1147 | 1104 | 1057 | 1059 | 1207 | 1153 |
| | | DetSAC | | 2237 | 1824 | 1691 | 1533 | 1338 | 2280 | 1822 | 1622 | 1482 | 1314 |
| | | SAC | | 2130 | 1888 | 1654 | 1485 | 1385 | 2239 | 1855 | 1704 | 1469 | 1257 |
| | | TD3 | | 2132 | 1922 | 1642 | 1796 | 1736 | 2037 | 2000 | 1786 | 1759 | 1694 |
| | Inverted-Pendulum-Swingup | DDPG | | 837 | 832 | 841 | 821 | 676 | 756 | 842 | 832 | 747 | 845 |
| | | DetSAC | | 839 | 887 | 887 | 888 | 889 | 792 | 886 | 888 | 889 | 889 |
| | | SAC | | 873 | 888 | 888 | 889 | 889 | 876 | 889 | 888 | 888 | 888 |
| | | TD3 | | 860 | 851 | 847 | 846 | 851 | 854 | 852 | 869 | 875 | 871 |
| | Mountain-Car | DDPG | | -0 | 70 | 74 | 36 | 45 | -0 | 74 | 74 | 74 | 45 |
| | | DetSAC | | 4 | 76 | 94 | 95 | 95 | 5 | 52 | 94 | 95 | 95 |
| | | SAC | | 9 | 85 | 94 | 95 | 95 | 4 | 85 | 95 | 95 | 95 |
| | | TD3 | | -0 | 5 | 52 | 94 | 94 | -0 | 94 | 79 | 84 | 84 |
| | Reacher | DDPG | | 16 | 16 | 15 | 15 | 15 | 16 | 16 | 15 | 15 | 15 |
| | | DetSAC | | 18 | 17 | 17 | 17 | 17 | 19 | 17 | 17 | 17 | 17 |
| | | SAC | | 18 | 17 | 17 | 17 | 16 | 18 | 18 | 17 | 17 | 17 |
| | | TD3 | | 17 | 19 | 19 | 19 | 19 | 17 | 17 | 16 | 16 | 16 |
| | Walker2D | DDPG | | 905 | 792 | 659 | 426 | 341 | 792 | 885 | 676 | 457 | 324 |
| | | DetSAC | | 1701 | 1371 | 986 | 891 | 789 | 1526 | 1201 | 923 | 914 | 767 |
| | | SAC | | 1642 | 1274 | 1018 | 882 | 837 | 1676 | 1413 | 1148 | 850 | 805 |
| | | TD3 | | 1759 | 1622 | 1294 | 1196 | 879 | 1842 | 1693 | 1611 | 1607 | 1630 |

Table I.3: This table shows the final evaluation return (mean over last five percent of training) for each configuration under the Logistic scheduler regime. The mean across all 20 runs is reported.

| | | | P | | | | | | | | | |
| Scheduler | Environment | Scale Type Algorithm | 0.1 Gauss | 0.5 Gauss | 0.9 Gauss | 1.3 Gauss | 1.7 Gauss | 0.1 OU | 0.5 OU | 0.9 OU | 1.3 OU | 1.7 OU |
|---|---|---|---|---|---|---|---|---|---|---|---|---|
| Constant | Half-Cheetah | DDPG | 192 | 218 | 343 | 322 | 259 | 174 | 204 | 254 | 238 | 249 |
| | | DetSAC | 1148 | 1272 | 743 | 594 | 577 | 1026 | 1070 | 869 | 690 | 651 |
| | | SAC | 1109 | 1413 | 567 | 437 | 354 | 1158 | 1102 | 848 | 676 | 621 |
| | | TD3 | 1284 | 1179 | 885 | 680 | 603 | 1053 | 1052 | 760 | 770 | 626 |
| | Hopper | DDPG | 950 | 498 | 343 | 288 | 222 | 946 | 717 | 572 | 407 | 321 |
| | | DetSAC | 1957 | 1100 | 814 | 671 | 632 | 1927 | 1458 | 1205 | 903 | 817 |
| | | SAC | 1976 | 1108 | 813 | 746 | 690 | 2044 | 1464 | 1215 | 906 | 813 |
| | | TD3 | 1911 | 1547 | 1199 | 1005 | 804 | 1961 | 1575 | 1178 | 963 | 776 |
| | Inverted-Pendulum-Swingup | DDPG | 738 | 751 | 746 | 748 | 743 | 738 | 760 | 755 | 753 | 760 |
| | | DetSAC | 793 | 857 | 855 | 858 | 864 | 815 | 843 | 840 | 842 | 842 |
| | | SAC | 841 | 860 | 856 | 863 | 860 | 827 | 850 | 850 | 846 | 847 |
| | | TD3 | 703 | 763 | 748 | 761 | 749 | 705 | 758 | 749 | 755 | 751 |
| | Mountain-Car | DDPG | -0 | 44 | 62 | 59 | 65 | -0 | 87 | 80 | 23 | 53 |
| | | DetSAC | 1 | 6 | 15 | 23 | 41 | 4 | 52 | 73 | 78 | 80 |
| | | SAC | -5 | 3 | 17 | 23 | 24 | -1 | 51 | 68 | 72 | 73 |
| | | TD3 | 2 | 82 | 58 | 70 | 70 | -0 | 77 | 69 | 67 | 87 |
| | Reacher | DDPG | 14 | 15 | 15 | 15 | 15 | 14 | 12 | 11 | 10 | 9 |
| | | DetSAC | 17 | 18 | 17 | 17 | 18 | 16 | 15 | 13 | 11 | 10 |
| | | SAC | 16 | 16 | 16 | 14 | 13 | 16 | 15 | 13 | 11 | 9 |
| | | TD3 | 15 | 12 | 11 | 10 | 9 | 14 | 12 | 11 | 10 | 9 |
| | Walker2D | DDPG | 484 | 367 | 280 | 243 | 239 | 498 | 455 | 327 | 336 | 346 |
| | | DetSAC | 1324 | 1034 | 448 | 301 | 293 | 1175 | 846 | 674 | 625 | 594 |
| | | SAC | 1413 | 1103 | 406 | 369 | 367 | 1290 | 824 | 658 | 600 | 588 |
| | | TD3 | 1504 | 1419 | 1098 | 842 | 730 | 1491 | 1440 | 1137 | 885 | 717 |
| Linear | Half-Cheetah | DDPG | 172 | 178 | 187 | 177 | 163 | 187 | 162 | 165 | 169 | 167 |
| | | DetSAC | 1230 | 1923 | 913 | 743 | 707 | 1030 | 1230 | 1186 | 907 | 905 |
| | | SAC | 1248 | 1840 | 966 | 654 | 627 | 1435 | 1346 | 1260 | 958 | 899 |
| | | TD3 | 1300 | 1727 | 1360 | 1099 | 1076 | 1292 | 1288 | 1243 | 1086 | 763 |
| | Hopper | DDPG | 896 | 769 | 607 | 527 | 498 | 877 | 774 | 696 | 594 | 546 |
| | | DetSAC | 2076 | 1551 | 1302 | 1174 | 977 | 2035 | 1667 | 1356 | 1131 | 969 |
| | | SAC | 1966 | 1554 | 783 | 713 | 701 | 1977 | 1601 | 1346 | 1117 | 999 |
| | | TD3 | 1772 | 1602 | 996 | 854 | 920 | 1854 | 1688 | 1452 | 1315 | 1177 |
| | Inverted-Pendulum-Swingup | DDPG | 713 | 749 | 757 | 756 | 748 | 729 | 753 | 755 | 751 | 671 |
| | | DetSAC | 818 | 861 | 857 | 857 | 859 | 774 | 844 | 838 | 840 | 840 |
| | | SAC | 850 | 860 | 857 | 858 | 862 | 835 | 847 | 849 | 848 | 848 |
| | | TD3 | 696 | 748 | 746 | 745 | 740 | 720 | 746 | 743 | 755 | 747 |
| | Mountain-Car | DDPG | -0 | 46 | 39 | 28 | 60 | -0 | 64 | 52 | 24 | 40 |
| | | DetSAC | 2 | -0 | 2 | 16 | 23 | 1 | 23 | 75 | 79 | 77 |
| | | SAC | -7 | -3 | -1 | 16 | 7 | -3 | 45 | 59 | 68 | 73 |
| | | TD3 | -0 | -0 | 33 | 46 | 73 | -0 | 69 | 68 | 69 | 68 |
| | Reacher | DDPG | 14 | 14 | 14 | 13 | 13 | 14 | 12 | 12 | 12 | 11 |
| | | DetSAC | 17 | 18 | 18 | 18 | 19 | 16 | 15 | 14 | 13 | 11 |
| | | SAC | 16 | 17 | 17 | 17 | 17 | 16 | 15 | 14 | 13 | 11 |
| | | TD3 | 15 | 17 | 17 | 17 | 17 | 14 | 12 | 12 | 11 | 11 |
| | Walker2D | DDPG | 484 | 467 | 372 | 293 | 256 | 509 | 495 | 371 | 331 | 272 |
| | | DetSAC | 1233 | 976 | 800 | 742 | 705 | 1206 | 973 | 860 | 757 | 689 |
| | | SAC | 1235 | 1291 | 622 | 510 | 491 | 1391 | 972 | 852 | 713 | 742 |
| | | TD3 | 1477 | 1384 | 918 | 665 | 612 | 1457 | 1397 | 1273 | 1176 | 1072 |
| Logistic | Half-Cheetah | DDPG | 175 | 168 | 166 | 166 | 162 | 182 | 177 | 160 | 167 | 165 |
| | | DetSAC | 1059 | 1358 | 1094 | 875 | 781 | 1065 | 1350 | 1122 | 967 | 768 |
| | | SAC | 1269 | 1433 | 1094 | 943 | 822 | 1329 | 1418 | 1115 | 912 | 842 |
| | | TD3 | 1224 | 1700 | 1292 | 943 | 931 | 1260 | 1234 | 1126 | 1109 | 915 |
| | Hopper | DDPG | 874 | 874 | 727 | 662 | 586 | 932 | 835 | 713 | 659 | 589 |
| | | DetSAC | 2064 | 1543 | 1361 | 1139 | 988 | 2027 | 1586 | 1268 | 1092 | 962 |
| | | SAC | 1911 | 1533 | 1306 | 1096 | 1000 | 2057 | 1589 | 1322 | 1073 | 941 |
| | | TD3 | 1827 | 1337 | 845 | 932 | 939 | 1803 | 1650 | 1358 | 1220 | 1104 |
| | Inverted-Pendulum-Swingup | DDPG | 726 | 756 | 750 | 757 | 589 | 656 | 748 | 759 | 676 | 739 |
| | | DetSAC | 794 | 841 | 838 | 842 | 843 | 777 | 846 | 845 | 843 | 840 |
| | | SAC | 843 | 850 | 849 | 849 | 845 | 838 | 856 | 852 | 848 | 840 |
| | | TD3 | 717 | 753 | 756 | 753 | 750 | 712 | 752 | 751 | 746 | 740 |
| | Mountain-Car | DDPG | -0 | 66 | 70 | 32 | 42 | -0 | 66 | 72 | 70 | 42 |
| | | DetSAC | 0 | 51 | 73 | 76 | 79 | 2 | 37 | 75 | 79 | 79 |
| | | SAC | -0 | 52 | 62 | 68 | 71 | -4 | 48 | 72 | 70 | 72 |
| | | TD3 | -0 | 3 | 39 | 71 | 81 | -0 | 81 | 71 | 80 | 80 |
| | Reacher | DDPG | 14 | 12 | 11 | 11 | 10 | 14 | 12 | 12 | 11 | 10 |
| | | DetSAC | 16 | 14 | 13 | 12 | 11 | 16 | 15 | 13 | 11 | 11 |
| | | SAC | 16 | 15 | 14 | 12 | 11 | 16 | 15 | 14 | 11 | 10 |
| | | TD3 | 15 | 17 | 17 | 17 | 17 | 15 | 13 | 12 | 11 | 11 |
| | Walker2D | DDPG | 508 | 468 | 352 | 269 | 237 | 498 | 462 | 350 | 267 | 242 |
| | | DetSAC | 1266 | 992 | 751 | 686 | 642 | 1146 | 922 | 732 | 699 | 650 |
| | | SAC | 1232 | 955 | 766 | 672 | 652 | 1237 | 1022 | 795 | 680 | 636 |
| | | TD3 | 1442 | 1313 | 672 | 608 | 476 | 1460 | 1326 | 1154 | 1000 | 982 |

Table I.4: This table shows the performance P as the mean across the 20 different seeds.

# J Hyperparameters

| Environment | Walker2D | Inverted-Pendulum-Swingup | Hopper | Mountain-Car | Half-Cheetah | Reacher |
|---|---|---|---|---|---|---|
| env_wrapper | **TimeFeature Wrapper** | | **TimeFeature Wrapper** | | **TimeFeature Wrapper** | |
| gamma | 0.99 | 0.99 | 0.99 | 0.99 | 1 | 0.99 |
| buffer_size | 1000000 | 1000000 | 1000000 | **50000** | 1000000 | 1000000 |
| learning_starts | **1000** | **1000** | **1000** | **0** | **10000** | **1000** |
| gradient_steps | **-1** | **-1** | **-1** | 1 | **-1** | **-1** |
| train_freq | **(1, 'episode')** | **(1, 'episode')** | **(1, 'episode')** | 1 | **(1, 'episode')** | **(1, 'episode')** |
| learning_rate | 0.0003 | 0.0003 | 0.0003 | 0.0003 | 0.0003 | 0.0003 |
| timesteps | **2000000** | **1000000** | **2000000** | 60000 | **2000000** | 1000000 |
| ID | **Walker2DBullet Env-v0** | **Inverted Pendulum SwingupPyBullet Env-v0** | **HopperPyBullet Env-v0** | **MountainCar Continuous-v0** | **HalfCheetah PyBulletEnv-v0** | **ReacherPyBullet Env-v0** |
| batch_size | 256 | **64** | 256 | **64** | 256 | **64** |
| ent_coef | **0** | **0** | **0** | auto | auto | **0** |
| tau | 0.005 | 0.005 | 0.005 | 0.005 | **0** | 0.005 |

a SAC/DetSAC Hyperparameters

| Environment | Walker2D | Inverted-Pendulum-Swingup | Hopper | Mountain-Car | Half-Cheetah | Reacher |
|---|---|---|---|---|---|---|
| env_wrapper | **TimeFeature Wrapper** | **TimeFeature Wrapper** | **TimeFeature Wrapper** | | **TimeFeature Wrapper** | **TimeFeature Wrapper** |
| gamma | **1** | **1** | **1** | 0.99 | **1** | **1** |
| buffer_size | **200000** | **200000** | **200000** | 1000000 | **200000** | **200000** |
| learning_starts | **10000** | **10000** | **10000** | 100 | **10000** | **10000** |
| gradient_steps | -1 | -1 | -1 | -1 | -1 | -1 |
| train_freq | (1, 'episode') | (1, 'episode') | (1, 'episode') | (1, 'episode') | (1, 'episode') | (1, 'episode') |
| learning_rate | 0.001 | 0.001 | 0.001 | 0.001 | 0.001 | 0.001 |
| policy_kwargs | **{'net_arch': [400, 300]}** | **{'net_arch': [400, 300]}** | **{'net_arch': [400, 300]}** | None | **{'net_arch': [400, 300]}** | **{'net_arch': [400, 300]}** |
| timesteps | **1000000** | **300000** | **1000000** | 300000 | **1000000** | **300000** |
| ID | **Walker2DBullet Env-v0** | **Inverted Pendulum SwingupPyBullet Env-v0** | **HopperPyBullet Env-v0** | **MountainCar Continuous-v0** | **HalfCheetah PyBulletEnv-v0** | **ReacherPyBullet Env-v0** |

b TD3 Hyperparameters

| Environment | Walker2D | Inverted-Pendulum-Swingup | Hopper | Mountain-Car | Half-Cheetah | Reacher |
|---|---|---|---|---|---|---|
| env_wrapper | **TimeFeature Wrapper** | **TimeFeature Wrapper** | **TimeFeature Wrapper** | | **TimeFeature Wrapper** | **TimeFeature Wrapper** |
| gamma | **1** | **1** | **1** | 0.99 | **1** | **1** |
| buffer_size | 1000000 | **200000** | 1000000 | 1000000 | **200000** | **200000** |
| learning_starts | **10000** | **10000** | **10000** | 100 | **10000** | **10000** |
| gradient_steps | -1 | -1 | -1 | -1 | -1 | -1 |
| train_freq | (1, 'episode') | (1, 'episode') | (1, 'episode') | (1, 'episode') | (1, 'episode') | 1 |
| learning_rate | **0.0007** | 0.001 | **0.0007** | 0.001 | 0.001 | 0.001 |
| policy_kwargs | **{'net_arch': [400, 300]}** | **{'net_arch': [400, 300]}** | **{'net_arch': [400, 300]}** | None | **{'net_arch': [400, 300]}** | **{'net_arch': [400, 300]}** |
| timesteps | **1000000** | **300000** | **1000000** | 300000 | **1000000** | **300000** |
| ID | **Walker2DBullet Env-v0** | **Inverted Pendulum SwingupPyBullet Env-v0** | **HopperPyBullet Env-v0** | **MountainCar Continuous-v0** | **HalfCheetah PyBulletEnv-v0** | **ReacherPyBullet Env-v0** |
| batch_size | **256** | 100 | **256** | 100 | 100 | 100 |

c DDPG Hyperparameters

Table J.1: Hyperparameters for SAC, TD3 and DDPG are taken from (Raffin, 2020) or left at default values defined in (Raffin et al., 2021a).

# K  Environment Limits

| Environment | MountainCarContinuous-v0 | InvertedPendulumSwingupPyBulletEnv-v0 | ReacherPyBulletEnv-v0 | HopperPyBulletEnv-v0 | Walker2DBulletEnv-v0 | HalfCheetahPyBulletEnv-v0 |
|---|---|---|---|---|---|---|
| $s^{(0)}$ | -1.2000…0.6000 | -1.0993…1.0931 | -0.2700…0.2700 | -1.2433…0.8614 | -1.2316…0.1270 | -0.6542…0.5536 |
| $s^{(1)}$ | -0.0700…0.0700 | -6.1276…6.0216 | -0.2700…0.2700 | -0.0000…0.0000 | -0.0000…0.0000 | -0.0000…0.0000 |
| $s^{(2)}$ | | -1.0000…1.0000 | -0.4799…0.4798 | -1.0000…1.0000 | -1.0000…1.0000 | -1.0000…1.0000 |
| $s^{(3)}$ | | -1.0000…1.0000 | -0.4799…0.4795 | -5.0000…3.4373 | -3.5129…1.8573 | -1.8748…2.0801 |
| $s^{(4)}$ | | -21.9001…21.2146 | -1.0000…1.0000 | -0.0000…0.0000 | -0.0000…0.0000 | -0.0000…0.0000 |
| $s^{(5)}$ | | | -1.0000…1.0000 | -5.0000…1.6368 | -3.6400…0.7000 | -1.9558…1.3548 |
| $s^{(6)}$ | | | -10.0000…10.0000 | -3.1416…3.1416 | -3.1416…0.0000 | -3.1416…0.0000 |
| $s^{(7)}$ | | | -1.2745…1.2701 | -1.5708…1.5342 | -1.5708…1.0625 | -1.5708…1.0959 |
| $s^{(8)}$ | | | -10.0000…10.0000 | -1.3921…2.1682 | -2.2274…1.5482 | -5.0000…1.1894 |
| $s^{(9)}$ | | | | -5.0000…5.0000 | -5.0000…4.6218 | -3.6561…3.8614 |
| $s^{(10)}$ | | | | -1.3917…1.8215 | -1.5703…1.8112 | -4.8688…2.1159 |
| $s^{(11)}$ | | | | -5.0000…5.0000 | -4.2228…4.3884 | -5.0000…3.7174 |
| $s^{(12)}$ | | | | -3.2458…2.1586 | -3.7981…1.5704 | -3.7094…4.0843 |
| $s^{(13)}$ | | | | -5.0000…5.0000 | -4.9543…3.1231 | -5.0000…5.0000 |
| $s^{(14)}$ | | | | -0.0000…1.0000 | -2.8370…1.2062 | -2.7263…1.6122 |
| $s^{(15)}$ | | | | | -4.9879…4.1318 | -5.0000…4.6168 |
| $s^{(16)}$ | | | | | -1.7225…1.7902 | -5.0000…3.2957 |
| $s^{(17)}$ | | | | | -3.7233…4.2734 | -5.0000…5.0000 |
| $s^{(18)}$ | | | | | -4.0686…1.5315 | -3.9515…3.3214 |
| $s^{(19)}$ | | | | | -5.0000…2.7369 | -5.0000…5.0000 |
| $s^{(20)}$ | | | | | -0.0000…1.0000 | -0.0000…1.0000 |
| $s^{(21)}$ | | | | | -0.0000…1.0000 | -0.0000…1.0000 |
| $s^{(22)}$ | | | | | | -0.0000…1.0000 |
| $s^{(23)}$ | | | | | | -0.0000…1.0000 |
| $s^{(24)}$ | | | | | | -0.0000…1.0000 |
| $s^{(25)}$ | | | | | | -0.0000…1.0000 |

Table K.1: The calculation of $X_{\mathcal{U}\text{rel}}$ requires defined state space limits for each environments. However, some environments define the limits as $(-\infty, \infty)$. In these cases we collected state space samples and defined the limits empirically.

