# OpenReview forum: "Action Noise in Off-Policy Deep Reinforcement Learning: Impact on Exploration and Performance"
_TMLR — Accepted by TMLR_

### Review · Reviewer_7knq · 2022-09-15

**Summary Of Contributions:**

The authors compare the role of two popular noise processes (Gaussian / O-U process) employed for exploration in DeepRL continuous control work. They evaluate different ways of setting the hyperparameters and schedule for these noise processes in 6 Mujoco domain and 4 different algorithms. They conclude that no single setting of the parameters leads to better state-space coverage or better performance across domains.


**Requested Changes:**

The paper relies on published codebases to comment about what has been explored and studied in the past (e.g. end of page 2). Please take the published literature into account rather than only available online code for comments about existing work. Case in point, reducing randomness over the course of learning is common both in practical algorithms and in theoretical RL work.

Can the returns be provided in Table 2, it would be more interpretable than the statistics to give a notion of the performance gaps. In general, I found the large number of statistical tests to be distracting.



**Strengths And Weaknesses:**

Strengths:

The authors motivate their investigation with a clear example in Mountain Car. After selecting the scope of their investigation (what metrics/domain/algorithms), they do an honest empirical work to analyze their findings.


Weaknesses:

The paper puts a lot of emphasis on state-space coverage to measure exploration efficacy, with a particular metric based on the divergence of a state distribution from a uniform distribution. It’s not clear to me that increasing that coverage is desirable by itself, and it also ignores the distribution over actions which is important for driving policy improvements.

A lot of space in the paper is given to description of standard methods (e.g. Eq 10-15, all of Appendix D) while comparatively less attention is given to details of the RL experimental setup. Section 3.5 describes what data is collected, but it’s not clear how that maps to the evaluation returns P and exploratory returns R for example.

Overall the empirical results are not particularly enlightening. Fig 5 or 7 for example illustrates that there is really little difference in aggregate between the two noise processes across the different metrics that were chosen (even if the means are statistically different, the difference is minor). It’s not clear what to take away from the paper and the results. The finding that the best noise process depends on the domain and the algorithm is not surprising in itself, and the generality of the heuristics in Section 5 seem rather limited (it’s based on 6 toy domains and relies on vague conditions like “reward is misleading”). I have trouble seeing these results generalize to more complex settings, and for this reason I don’t think the community will benefit from these findings - even if they are valid.

---

### Review · Reviewer_oEpB · 2022-09-21

**Summary Of Contributions:**

## Summary
This paper provides an analysis of the effects of using action noise for exploration in deep actor-critic algorithms. In particular, the paper studies how adding Gaussian and OU noise to actions affects multiple facets of Soft Actor-Critic (SAC), Deep Deterministic Policy Gradient (DDPG), and Twin Delayed Deep Deterministic Policy Gradient (TD3). These facets particularly are the return achieved and the state space coverage of the exploitation policy (where action noise is not used) and the exploration policy (where action noise is used). The overall conclusions are that action noise does affect the performance and state-space coverage attained by these algorithms, but that the type of action noise and the scale of action noise should be tuned for each environment. Furthermore, reducing the action noise throughout training seems to improve performance.  The paper introduces a new method to measure state-space coverage which addresses issues with current methods of measuring state space coverage. In particular, this novel method is less sensitive to data points lying along the state space boundaries.


**Broader Impact Concerns:**

No broader impact concerns.

**Requested Changes:**

The paper should comment on whether or not the assumptions of each statistical significance test is satisfied. If the assumptions are not satisfied, then the experiments should instead use a weaker statistical significance test for which the assumptions are satisfied. This detail is critical to securing my recommendation for acceptance.

Tuning the other hyperparameters for each noise type, scale, and scheduler would significantly strengthen the work and be beneficial to the RL research community as a whole. Even if only a subset of all other hyperparameters were tuned, this would strengthen the work greatly.
This could even be accomplished by first tuning **across all environments** over a small number of runs, then evaluating the tuned hyperparameters across all environments using a larger number of runs.

**Strengths And Weaknesses:**

## Main Argument
Overall, the paper provides a much needed analysis on the effects of action noise in deep actor-critic algorithms. The effects of both Gaussian and OU noise on deep actor-critic algorithms have not been thoroughly studied, and this paper provides a comprehensive analysis of this topic.

Unfortunately, a number of empirical design choices cause me to question the overall results of the paper.

First, the paper uses multiple different types of statistical significance tests. These tests each have their own assumptions on the form of the data to which they are applied. The paper does not comment on whether or not such assumptions are satisfied. For example, the t-test assumes data from a normal distribution (otherwise a normal sampling distribution for the mean, but this would be difficult to verify), yet the paper does not verify if this assumption is satisfied. Similar issues exist for the other tests which are utilized in the paper. Does the data satisfy the assumptions for each statistical significance test? If not, then the reported results of such tests may be overly optimistic.

The empirical study uses hyperparameters tuned for a specific action noise scale and schedule. Then re-uses these same hyperparameters when changing the action noise scale and schedule. For example in the Stable Baselines codebase, TD3 and DDPG use the hyperparameters for TD3 reported in [1], which were tuned for Gaussian noise with noise scale $\sigma$ and a constant noise scheduler.  All other hyperparameters of TD3 have been tuned to work well with this noise type, scale, and schedule.

In the experiments outlined in this paper, hyperparameters are not re-tuned after changing the noise type, scale, or schedule. Instead, the default hyperparameters tuned for the original noise type, scale, and schedule are re-used, but these hyperparameters may no longer be appropriate for the altered action noise.  For example, since the step-sizes were tuned for TD3 using Gaussian noise for exploration, simply substituting OU noise for Gaussian noise likely results in a pessimistic estimate of performance for OU noise in TD3.

This could bias results, giving the noise type, scale, and schedule that was originally tuned for an unfair advantage. For example, in Figure 5b, Gaussian noise results in higher (exploratory) rewards than does OU noise. But since the hyperparameters used were tuned for Gaussian noise, it is likely the case that Gaussian noise had an advantage over OU noise in this experiment. Had the hyperparameters additionally been tuned for OU noise, perhaps different conclusions would have been drawn. I understand that we cannot perform a systematic sweep over all possible hyperparameters, as this would be prohibitively expensive.  Nevertheless, this is a significant limitation of this work, and the work would be greatly improved with a more thorough consideration of hyperparameter selection.

Section 3.6.1, provides an experiment on each method for measuring state space coverage. A number of data points are sampled from a 25-dimensional distribution. Then, the scale of these data points are changed to vary the amount of state space coverage, including causing many data points to lie on the boundaries of the state space. Although this is a useful experiment, only 10 random seeds were used to generate the results. From my understanding, these are quite cheap experiments to run, and so the choice to use so few random seeds confused me. Furthermore, why was only the mean plotted and not a confidence interval? Perhaps there is some part of the algorithm that I am mis-understanding.

In appendix G, the figure states that the random seed used for each experiment was drawn independently and randomly. What were the bounds and distribution from which these random seeds were drawn? Were the seeds sampled with or without replacement? Why was the same random seed not used for each algorithm for each separate experimental trial? This would ensure each algorithm saw the same initial weights, same environmental starting states, etc.

[1] Scott Fujimoto, Herke van Hoof, David Meger. Addressing Function Approximation Error in Actor-Critic Methods. 2018.

## Small things which did not affect the scoring of the paper:

- Appendix E: The text in the figure is incredibly small, making it hard to read. Furthermore, the section would benefit from some analysis of the figure.
- Appendix G: Table G.1 is unnecessary and doesn't convey any additional information. It suffices to say "each experiment used 20 runs"
- Section 3.3: "See Brockman et al. (2016), Coumans and Bai (2016), and Ellenberger (2018) for further details.". You cannot see the references, and the sentence would read better as "Brockman et al. (2016), Coumans and Bai (2016), and Ellenberger (2018) provide further details."
- Section 2 (Related Work) states that deterministic policies can **only** be used in the off-policy setting. This is incorrect. A deterministic policy can be used in the on-policy setting, but is discouraged due to a lack of exploration.

---

### Review · Reviewer_jTC2 · 2022-09-22

**Summary Of Contributions:**

The paper conducts a thorough empirical investigation of the effects of various configurations of action noise (scale, type, reduction schedule) on performance for several deep RL algorithms and environments in the continuous control setting. The paper also proposes a measure of state space coverage with desirable properties, and gives two estimators that implement the proposed measure. In addition, the paper gives several helpful recommendations for choosing action noise parameters, that are rooted in the experiment results.

**Broader Impact Concerns:**

Doesn't seem applicable.

**Requested Changes:**

**Critical changes:**
1. The Related Work section doesn't actually mention any papers that empirically examine the effects of action noise on algorithm performance, even for different algorithms or settings (like epsilon-greedy in the finite action setting, which is instead discussed in the introduction). It actually seems more like a background section that introduces the concepts that the paper builds on. I recommend changing the name of the Related Work section to "Background", and including a Related Work section after the introduction that summarizes the works that are most closely related to the main point of the paper (i.e., any papers that study action noise, even in another setting or algorithm). This section should also compare and contrast the findings of related works to better position this paper's contributions within the literature. For example, do the findings of the current paper echo findings from the finite-action setting, or are there important/surprising differences?
1. It wasn't clear to me what "boundary artifacts" meant until I read section 3.6. Then it wasn't clear to me how they are meaningfully different from "outliers"; whether the space is bounded or not, values far from the others can disproportionately affect certain estimators. Then after seeing Figure 4, I think I understood better what is meant by "boundary artifacts" and how they could be different from outliers (e.g., the clipping process causing samples to concentrate around the boundaries instead of the interior of the space). I would suggest either defining "boundary artifacts", giving examples, or coming up with a name that better communicates the cause of the phenomena (i.e., "clipping artifacts"). Earlier in the paper when it's first used would be ideal.
1. The caption for figure 4 (and later figures) repeats details that were stated in the main text just before it. I would suggest moving all details for all figures to the main text, and having more succinct captions that include details related to the specific figure (e.g., "best performance in bold", "shaded regions are standard error", etc.), and a summary of the main findings for that figure (similar to the captions of figures 4(a) and 4(b), which I liked a lot).

**Changes that would strengthen the work in my view:**
1. In the paragraph following "Findings", the paper states "Larger noise scales tend to increase state-space coverage, but for the majority of our investigated environments increasing the state-space coverage is not beneficial." It seems like increasing the state-space coverage specifically by increasing the action noise scale is not beneficial, but more principled ways of increasing the state-space coverage could still be beneficial. It would be good to clarify this statement.
1. Should "In case of action noise" be "In the case of action noise"?
1. It would be good to be very clear that the version of Mountain Car described in the paper is not the usual version from OpenAI Gym.
1. The square root symbol in equation 3 only extends to the left parenthesis.
1. Why sample noise from an unbounded Gaussian and clip values instead of sampling from a truncated Gaussian distribution directly? I understand that this setup is something others have done in the past, but it would be good to justify building on top of a bad legacy, perhaps by appealing to practitioners looking to use and understand existing methods.
1. It's good to use the past tense to communicate to the reader what was actually done in the experiments, and save the present tense for statements that are generally true and conclusions that are being drawn from the experiments.
1. The first sentence of section 3.5 is a little confusing as is. It says "we...evaluate the exploration and learned policy performance once for each...segment", which made me think one rollout was performed, but then in the next sentence the paper states that either 100 episodes or 10000 steps were performed. I would recommend rephrasing the first sentence to make it clearer, maybe by just removing "once".
1. The discussion at the bottom of page 8 feels like it jumps back and forth a few times describing the kNN and NNR estimators. It might be better to introduce both estimators in one sentence, fully describe one estimator (1 paragraph), and then fully describe the other (another paragraph). That way, when comparisons between the estimators are made, the reader already knows what they need to know to understand the comparison.
1. It would be good to explain why C_L and C_U were chosen how they were (1/n and n/1, respectively); as it is these choices don't make a ton of intuitive sense to me.
1. It wasn't clear to me what was meant by "the environment acts as an integrator over the actions." It could be good to elaborate.
1. In the caption of Table 5, the first letter of the second sentence should be capitalized.

**Questions:**
1. The description of the experiments (top of page 7) says they "would" amount to 244 node-days with the specified hardware setup. It's always better to describe what was actually done. Did the experiments take 244 days to run, or were they run on a cluster?
1. Why add action noise to SAC? It already randomizes its actions by maximizing the entropy of the policy in addition to maximizing value, which seems like a better way of achieving the same goal as adding action noise.
Maximizing policy entropy as well as value allows the agent to act randomly when it makes sense, but act less randomly when it would be harmful. Adding action noise seems redundant; it would make more sense to increase the entropy hyperparameter to cause the agent to act more randomly.
1. Why was the evaluation for each segment done the (seemingly complex) way it is (i.e., 100 episodes or 10000 steps, whichever is reached first, using only complete episodes)? Is it due to the nature of the environments (e.g., some episodic, some continuing)? It's always better to give exactly equal amounts of experience to each algorithm to aid comparisons; I can't tell if that is being done here or not. At the very least it would be good to justify or explain why this evaluation methodology was chosen.
1. I like the symmetric KL-divergence-based measure of state space coverage; it seems much closer to what we actually want, with reasonable added complexity. Have you considered something like the earth mover's distance/Wasserstein metric (might be overly-complicated/overkill)?

**Misc comments:**
1. I greatly appreciated the explanation justifying the choice of noise scale parameter values.
1. Thank you for explaining the reasoning behind using the rollouts instead of the replay buffer to measure returns and state space coverage; it was very helpful.
1. Likewise, Figure 3 was very helpful for getting an intuitive understanding of each of the measures of state space coverage. Thank you for taking the time and effort to create and include it.
1. Again, thank you for justifying/explaining the logic behind the KL-divergence based measure.
1. I liked how the recommendations are stated (pages 15-16), and then explained. This should be really easy for readers to understand, and for practitioners to implement.


**Strengths And Weaknesses:**

**Strengths:**
1. Well written and mostly clear. Very easy to read!
1. Empirical investigation is thorough and seems principled.
1. Design decisions are mostly explained and justified.

**Weaknesses:**
1. Is using action noise for exploration even a good idea? It's unlikely to be the best solution to the exploration problem and is more of a stand-in for principled exploration strategies. On the other hand, it is commonly used and strongly affects the performance of the algorithms that use it, so might be worth exploring.
1. The findings of the paper are not necessarily surprising, and some of the recommendations seem to be common sense. Hyperparameters are often already set on a per-environment basis. Reducing exploration (in this case implemented as action noise) over time is common, although apparently not in the existing software packages cited in the paper. Likewise, it's unsurprising that the scale of the noise has the largest impact on performance of the factors considered. However, the paper's thorough empirical investigation gives concrete evidence to support this folk wisdom, and is a helpful contribution, especially for practitioners looking to use the studied algorithms on new environments.

---

### Decision · Action_Editors · 2022-10-31

**Recommendation:** Accept as is

**Comment:**

All reviewers agreed on accept as is.

**Audience:**

Empirical insights into exploration in RL---so yes!

**Claims And Evidence:**

This paper clearly fits the bill for TMLR: well done, correct, claims backed by evidence, measured and the work is certainly of interest to people in the RL community. All the reviewers agreed.